# LATENT SAFETY-CONSTRAINED POLICY APPROACH FOR SAFE OFFLINE REINFORCEMENT LEARNING

**Prajwal Koirala, Zhanhong Jiang, Soumik Sarkar & Cody Fleming**
Iowa State University
Ames, Iowa, USA
`{prajwal,zhjiang,soumiks,flemingc}@iastate.edu`

## ABSTRACT

In safe offline reinforcement learning (RL), the objective is to develop a policy that maximizes cumulative rewards while strictly adhering to safety constraints, utilizing only offline data. Traditional methods often face difficulties in balancing these constraints, leading to either diminished performance or increased safety risks. We address these issues with a novel approach that begins by learning a conservatively safe policy through the use of Conditional Variational Autoencoders, which model the latent safety constraints. Subsequently, we frame this as a Constrained Reward-Return Maximization problem, wherein the policy aims to optimize rewards while complying with the inferred latent safety constraints. This is achieved by training an encoder with a reward-Advantage Weighted Regression objective within the latent constraint space. Our methodology is supported by theoretical analysis, including bounds on policy performance and sample complexity. Extensive empirical evaluation on benchmark datasets, including challenging autonomous driving scenarios, demonstrates that our approach not only maintains safety compliance but also excels in cumulative reward optimization, surpassing existing methods. Additional visualizations provide further insights into the effectiveness and underlying mechanisms of our approach. The code is available here.

## 1 INTRODUCTION

Although Reinforcement learning (RL) is a popular approach for decision-making and control applications across various domains, its deployment in industrial contexts is limited by safety concerns during the training phase. In traditional online RL, agents learn optimal policies through trial and error, interacting with their environments to maximize cumulative rewards. This process inherently involves exploration, which can lead to the agent encountering unsafe states and/or taking unsafe actions, posing substantial risks in industrial applications such as autonomous driving, robotics, and manufacturing systems (García & Fernández, 2015; Gu et al., 2022; Moldovan & Abbeel, 2012; Shen et al., 2014; Yang et al., 2020). The primary challenge lies in ensuring that the agent's learning process does not compromise safety, as failures during training can result in costly damages, operational disruptions, or even endanger human lives (Achiam et al., 2017; Stooke et al., 2020). To address these challenges, researchers have explored several approaches aimed at minimizing safety risks while maintaining the efficacy of RL algorithms.

One effective method to mitigate safety risks associated with training an agent is offline RL. In this paradigm, the focus shifts from active interaction with the environment to learning policies from a static dataset. This dataset comprises trajectory rollouts generated by an arbitrary behavior policy or multiple policies, collected beforehand. By leveraging this fixed dataset, offline RL eliminates the need for real-time data collection, thereby significantly reducing the risk of actually encountering unsafe states during the learning process. Training an agent with Offline RL, however, presents a unique set of challenges, primarily due to the issue of distribution shift (Levine et al., 2020; Tarasov et al., 2024; Fu et al., 2020). The static dataset may not fully represent the range of scenarios the agent will encounter in the real world, leading to potential mismatches between the training data and the learned policy. This discrepancy can result in suboptimal policy performance when deployed in real-world settings. Despite these challenges, offline RL remains a powerful tool for safely training RL agents, as it allows for policy evaluation and improvement without incurring the risks associated with live interactions.

Another critical approach to mitigating safety risks in RL is safe RL. Unlike traditional RL, where the primary objective is to maximize cumulative rewards, safe RL places a strong emphasis on producing actions that adhere to predefined safety constraints (Xu et al., 2022b; Chow et al., 2018). This involves integrating safety considerations directly into the RL framework, ensuring that the agent's behavior remains within acceptable safety bounds throughout the training and deployment phases. Safe RL methods often formulate the problem as a Constrained Markov Decision Process (CMDP), where the agent not only seeks to optimize its performance but also to satisfy safety constraints (Altman, 1998; 2021; Chow et al., 2018). This dual objective can be challenging to achieve, as it requires balancing the exploration needed for learning with the strict adherence to safety requirements.

In safe offline RL, the agent is tasked with learning a policy exclusively from pre-collected data while adhering to stringent safety constraints (Liu et al., 2023a; Le et al., 2019). It is common practice—and often necessary—to constrain the policy such that it selects actions not only within the support of the dataset but also in compliance with the safety constraints (Xu et al., 2022a). However, solving this problem is inherently difficult due to the trade-offs involved in enforcing constraints. Overly restrictive constraints can limit the agent's ability to explore potentially rewarding actions, leading to suboptimal policies that fail to optimize for cumulative rewards. On the other hand, overly relaxed constraints may allow the selection of out-of-distribution (OOD) actions, increasing the risk of violating safety constraints and leading to hazardous outcomes.

To address these challenges, we introduce Latent Safety-Prioritized Constraints (LSPC), a framework that leverages Conditional Variational Autoencoders (CVAEs) to model the distribution of safety constraints within a latent space. This allows the agent to operate within a learned safety-prioritized boundary while maintaining sufficient flexibility for reward optimization. By incorporating implicit Q-learning, our approach formulates the policy learning problem as a Constrained Reward-Return Maximization task, where the agent seeks to maximize cumulative rewards while adhering to the inferred safety constraints. This principled method ensures that the learned policy remains safe, even in offline settings where exploration is not possible. Our empirical results on standard safe offline RL benchmarks demonstrate that LSPC achieves superior performance compared to existing approaches, balancing safety and reward return in a manner that is suitable for deployment in critical, high-stakes environments. The key contributions of this paper are as follows:

- We propose LSPC, a novel framework to model safety constraints derived from the static dataset in a tractable and imposable form, thereby facilitating their integration into the policy learning process.
- We formulate the problem as Constrained Reward-Return Maximization, ensuring safety compliance while maximizing cumulative rewards, supported by theoretical bounds on policy performance and sample complexity.
- Through extensive empirical evaluations, we demonstrate that policies derived from LSPC significantly outperform existing methods in both safety and reward optimization, providing a robust solution for high-stakes environments.

## 2 PRELIMINARIES

### 2.1 SAFE OFFLINE RL

In Safe Reinforcement Learning (Safe RL) problems, the environment is defined as a Constrained Markov Decision Process (CMDP), represented by the tuple $\mathcal{M} = (\mathcal{S}, \mathcal{A}, \mathcal{P}, r, c, \gamma, \rho_0)$. Here, $\mathcal{S}$ represents the state space, $\mathcal{A}$ the action space, $\mathcal{P} : \mathcal{S} \times \mathcal{A} \times \mathcal{S} \to [0, 1]$ the state transition probability function, $r : \mathcal{S} \times \mathcal{A} \to \mathbb{R}$ the reward function, $c : \mathcal{S} \times \mathcal{A} \to [0, C_m]$ the cost function associated with constraint violations, $\gamma$ the discount factor, and $\rho_0$ the initial state distribution. $C_m$ is the maximum value for each immediate cost. We also assume that the maximum value of the immediate reward is $R_m$. The cost function penalizes transitions that violate safety constraints, and the objective is to ensure safety by keeping the cumulative cost under a predefined threshold, $\kappa$.

A trajectory $\tau = (s_0, a_0, r_0, c_0), (s_1, a_1, r_1, c_1), \ldots, (s_T, a_T, r_T, c_T)$ represents a sequence of states, actions, rewards, and costs over time. The discounted cumulative reward for a trajectory is defined as $R(\tau) = \sum_{t=0}^{T} \gamma^t r(s_t, a_t)$, and the discounted cumulative cost is $C(\tau) = \sum_{t=0}^{T} \gamma^t c(s_t, a_t)$. We then also define the stationary state-action distribution under the policy $\pi$ as $d^\pi(s, a) = (1 - \gamma) \sum_{t=0}^{T} \gamma^t p(s_t = s, a_t = a)$ and the stationary state distribution under the policy $\pi$ as $d^\pi(s) =$

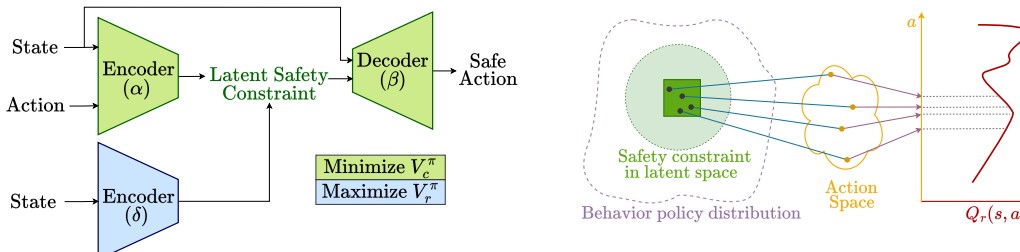

(a) Illustration of the architecture and training objective of proposed safe offline RL framework.

(b) Visual representation of action selection within the latent safety-constrained space.

Figure 1: Overview of the Proposed Method: A safe offline RL framework that balances safety and reward optimization. (a) The state-action pairs are encoded into a latent space by encoder $\alpha$, where the latent safety constraints are inferred. Decoder $\beta$ then uses these constraints to generate safe actions. Simultaneously, another encoder $\delta$ maximizes the reward signal to find actions that optimize performance while adhering to safety constraints. The process balances between minimizing the violation of safety constraints $V_c^\pi$ and maximizing reward returns $V_r^\pi$. (b) Actions from the latent space are subject to additional restriction to ensure they meet safety requirements before being optimized for reward return. The Q-function $Q_r(s, a)$ represents the expected reward return for each safe action for that state.

$(1 - \gamma) \sum_{t=0}^{T} \gamma^t p(s_t = s)$, where $p$ signifies the probability. The goal in safe RL is to learn a policy that maximizes the expected return in the CMDP while keeping the expected cost return below an allowable threshold. Thus, the safe RL problem is mathematically defined as follows:

$$\max_\pi \mathbb{E}_{\tau \sim \pi}[R(\tau)], \text{s.t.}, \mathbb{E}_{\tau \sim \pi}[C(\tau)] \leq \kappa \tag{1}$$

Offline Reinforcement Learning is a variant of RL in which the learning process is confined to a static dataset, eliminating any further interaction with the environment during training. The agent learns the policy from a pre-collected dataset $\mathcal{D} := (s, a, s', r, c)$, with both safe and unsafe trajectories. The value functions for both reward and cost can be defined in a unified way $V_h^\pi(s_0) = \mathbb{E}_{\tau \sim \pi}[\sum_{t=0}^{T} \gamma^t h_t | s_t = s_0], h \in \{r, c\}$. Denote by $\pi_b$ the unknown behavior policy induced by the dataset $\mathcal{D}$. We can now formulate safe offline RL as an optimization problem within the framework of a CMDP:

$$\max_\pi V_r^\pi(s), \ s.t., V_c^\pi(s) \leq \kappa; \ D_{KL}(\pi || \pi_b) \leq \varepsilon_1, \tag{2}$$

where $\pi_b$ governs the action distribution in the given dataset, and $\varepsilon_1$ is a divergence tolerance parameter. The function $D_{KL}(\pi || \pi_b)$ measures the divergence between the learned policy $\pi$ and the behavior policy $\pi_b$, typically using metrics such as Kullback-Leibler (KL) divergence or other statistical distance measures. These constraints ensure that the learned policy $\pi$ remains within the safety limits defined by the cost threshold $\kappa$ and does not deviate significantly from the behavior policy $\pi_b$, thereby mitigating the risks associated with out-of-distribution actions.

## 2.2 CONDITIONAL VARIATIONAL AUTOENCODER

A Conditional Variational Autoencoder (CVAE) is a generative model that extends the standard Variational Autoencoder (VAE) by conditioning the generation process on additional information (Doersch, 2016). Formally, the objective of the CVAE is to maximize the conditional likelihood $p_\theta(x|y)$ of the data $x$ given the condition $y$. To achieve this, CVAE introduces a latent variable $z$ and optimizes the variational evidence lower bound (ELBO) on the conditional log-likelihood:

$$\log p(x|y) \geq \mathbb{E}_{q(z|x,y)}[\log p(x|z,y)] - D_{KL}(q(z|x,y) || p(z|y)),$$

where $q(z|x, y)$ is the variational posterior, $p(x|z, y)$ is the likelihood, and $p(z|y)$ is the prior. The KL divergence term $D_{KL}(q(z|x,y) || p(z|y))$ regularizes the learned posterior to be close to the prior. During training, the CVAE jointly optimizes the encoder and decoder networks by minimizing the negative ELBO, which balances the reconstruction error and the regularization term. This approach allows the CVAE to generate new data samples $(x)$ from the distribution of the dataset conditioned on the variable $y$.

## 2.3 IMPLICIT Q LEARNING

Offline Reinforcement Learning (RL) algorithms often address covariate distribution shift by employing various strategies, such as using regularization methods to constrain the policy and/or the critic. In some cases, they avoid training the model outside the support of the distribution of the dataset altogether. Implicit Q-learning (IQL) incorporates an additional state-value network to prevent querying out-of-distribution actions during the training of the Q-network (Kostrikov et al., 2021b). Notably, training IQL does not require sampling actions from a policy but can be performed solely using actions from the dataset. The typical losses for IQL are as follows:

$$\mathcal{L}_Q(\theta) = \mathbb{E}_{(s,a,s')\sim\mathcal{D}} \left[ \left(r + \gamma V_\phi(s') - Q_\theta(s,a)\right)^2 \right] \tag{3}$$

$$\mathcal{L}_V(\phi) = \mathbb{E}_{(s,a)\sim\mathcal{D}} \left[ L_\xi^2 \left( Q_\theta(s,a) - V_\phi(s) \right) \right] \tag{4}$$

The IQL framework involves training a value network, denoted as $V_\phi$, in addition to the Q-network, denoted as $Q_\theta$. The loss functions associated with training these critic networks are detailed in Eqs. 3 and 4. State value network ($V_\phi$) is trained with expectile regression objective and uses an asymmetric squared error loss function defined as $L_\xi^2(u) = |\xi - 1(u < 0)|u^2$, where $\xi \in (0.5, 1.0)$. With both Q and value networks in place, the policy can be trained or extracted using the advantage-weighted regression (AWR) method (Kostrikov et al., 2021b; Peng et al., 2019). This method uses the learned advantage function, which is derived from the Q and value networks, and is defined as $A(s,a) = Q_\theta(s,a) - V_\phi(s)$, to guide the policy training process effectively within the distributional constraints of the dataset.

IQL in the context of offline RL is focused on learning reward-value from the data and using reward-advantage weighted regression to extract a policy that predicts actions with higher expected reward-return. A similar method can be used to learn the cost-value from the offline data. We use the superscript $c$ to denote the cost and define the cost-advantage of an action taken at state $s$ as $A^c(s,a) = Q_\psi^c(s,a) - V_\eta^c(s)$ . A similar asymmetric loss can be used to learn the cost-value of a state to discourage the underestimation of $Q_\theta^c(s,a)$:

$$\mathcal{L}_Q^c(\psi) = \mathbb{E}_{(s,a,s')\sim\mathcal{D}} \left[ \left(c + \gamma V_\eta^c(s') - Q_\psi^c(s,a)\right)^2 \right] \tag{5}$$

$$\mathcal{L}_V^c(\eta) = \mathbb{E}_{(s,a)\sim\mathcal{D}} \left[ L_\xi^2 \left( V_\eta^c(s) - Q_\psi^c(s,a) \right) \right] \tag{6}$$

## 3 METHODOLOGY

In this section, we detail the formulation of the proposed Latent Safety Prioritized Constraints (LSPC), which we employ to formulate the constrained reward optimization problem. As depicted in figure 1, the framework consists of two key components: (1) the derivation of a conservatively safe policy with a Conditional Variational Autoencoder (CVAE), and (2) the maximization of reward returns subject to the safety constraints inferred within the latent space.

As we will see, the CVAE-based architecture encodes state-action pairs into a latent representation that prioritizes adherence to safety constraints while being trained to reconstruct *safe* actions. Moreover, it also serves as a parametric generative model to implicitly enforce the policy to output actions within the support of the dataset. By selecting the action most likely in the dataset $\mathcal{D}$, the generative model helps to minimize errors in value (both cost and reward) approximation during policy evaluation, which often arise in offline RL due to out-of-distribution (OOD) state-action pairs (Fujimoto et al., 2019; Zhou et al., 2021). This obviates the need of explicit constraint on the divergence between the learned policy ($\pi$) and behavior policy ($\pi_b$).

Thus, the CVAE plays an important role in enforcing the two constraints outlined in Equation 2, ensuring that the policy operates within the boundaries of safe behavior as defined within the dataset. The latent space inferred through this training encapsulates these constraints in a more tractable form derived solely from a static, cost-labeled dataset. For these reasons, the constraints imposed via the latent space in our method are referred to as Latent Safety-Prioritized Constraints (LSPC). Below, we derive two specific policies: LSPC-S, a conservative policy trained solely to predict actions within these safety constraints, and LSPC-O, which is optimized to predict high reward-value actions while still adhering to these constraints.

## 3.1 LEARNING CONSERVATIVELY SAFE POLICY

**Behavior Policy Modeling.** To derive a safe policy $\pi_s(a|s)$ (the subscript $s$ in $\pi_s$ is short for safe), we start by employing a conditional variational autoencoder (CVAE) to model the underlying behavior policy $\pi_b(a|s)$ in the dataset. The objective of the CVAE is to maximize the likelihood $\log \pi_b(a \mid s)$ by maximizing a variational lower bound formulated as:

$$\max_{\alpha,\beta} \log \pi_b(a \mid s) \geq \max_{\alpha,\beta} \mathbb{E}_{z \sim q_\alpha} \left[ \log p_\beta(a \mid s, z) \right] - D_{KL} \left[ q_\alpha(z \mid s, a) \parallel p(z \mid s, a) \right] \qquad (7)$$

where $z$ is the latent variable, $\alpha$ and $\beta$ are the parameters of the encoder and the decoder, respectively. A trained encoder $q_\alpha(z \mid s, a)$ encodes the state-action pair to a probability distribution in the latent space. A trained decoder $p_\beta(a \mid s, z)$ provides a mapping from the latent space to the action space, conditioned on the state. When the latent variable $z$ takes on values that are likely under the prior $p(z \mid s, a)$, which is modeled as a standard normal distribution $\mathcal{N}(0, 1)$, the decoder $p_\beta(a \mid s, z)$ is expected to generate actions that align closely to the behavior policy distribution $\pi_b(a \mid s)$.

**Safe Policy Derivation.** We define a safe policy as one that minimizes the expected cumulative discounted cost return over the course of interactions with the environment:

$$\pi_s = \arg\min_\pi \mathbb{E}_\pi \left[ \sum_{t=0}^{T} \gamma^t c(s_t, a_t), a_t \sim \pi(\cdot \mid s_t) \right] \qquad (8)$$

In off-policy actor-critic methods, safe policy extraction can be achieved using advantage-weighted regression (AWR). This approach reformulates the problem to maximize the expected log likelihood of actions weighted by their advantage over the cost. The safe policy $\pi_s$ can be derived as follows:

$$\pi_s = \arg\max_\pi \mathbb{E}_{(s,a) \sim \mathcal{D}} \left[ \exp \left( \lambda \left( V_\eta^c(s) - Q_\psi^c(s, a) \right) \right) \log \pi(a \mid s) \right], \qquad (9)$$

where $\lambda \in [0, \infty)$ is a hyperparameter in AWR called inverse temperature. This formulation is equivalent to maximizing the log likelihood ($\pi_b(a \mid s)$) of an action given a state, with weights assigned according to the advantage, thereby promoting actions that are expected to minimize the cumulative cost. In the CVAE framework, parameters of the safe policy can be derived as $\alpha^*, \beta^* = \arg\min_{\alpha,\beta} \mathcal{L}_{p,q}(\alpha, \beta)$, where

$$\mathcal{L}_{p,q}(\alpha, \beta) = \mathbb{E}_{(s,a) \sim \mathcal{D}} \left[ -\exp \left( \lambda \left( V_\eta^c(s) - Q_\psi^c(s, a) \right) \right) \left( \mathbb{E}_{z \sim q_\alpha} \left[ \log p_\beta(a \mid s, z) \right] \right. \right.$$
$$\left. \left. - D_{KL} \left[ q_\alpha(z \mid s, a) \parallel p(z \mid s, a) \right] \right) \right] \qquad (10)$$

The CVAE is trained to reconstruct safe actions conditioned on given states and actions from the static dataset, effectively capturing the distribution of a safe policy.

$$\pi_s(a \mid s) = \int_z p_\beta(a \mid s, z) p(z \mid s, a) \, dz = \mathbb{E}_{p(z \mid s, a)} [p_\beta(a \mid s, z)] \qquad (11)$$

For the values of $z$ that have a high probability under the prior $p(z \mid s, a)$, the decoder $p_\beta(a \mid s, z)$ is expected to generate actions with high likelihood under the policy distribution $\pi_s(a \mid s)$. Thus, in our framework we define $z$ as an LSPC constraint, whose distribution is modeled as a Gaussian with zero mean and unit variance, ensuring that the sampled actions align with the safe policy distribution in expectation. The latent safety constraint serves as a probabilistic embedding that make the safety constraints present in the static dataset more tractable. When sampling $z$ from the full prior $\mathcal{N}(0, 1)$, we refer to the resulting policy as the CVAE policy, which uses the decoder to generate 'safe' actions conditioned on the state. In practice, we restrict the sampling of $z$ to a high-probability region of the prior, leading to a safer policy due to this restriction. We refer to the resulting policy as LSPC-S, as it prioritizes safety during execution.

## 3.2 CONSTRAINED REWARD-RETURN MAXIMIZATION

The objective of this constrained optimization problem is to solve for a policy $\pi$ that selects action with the maximum $Q_r$ value while also respecting the latent safety constraint.

$$\pi(s) = \arg\max_a Q_r(s, a), a \sim p_\beta(\cdot \mid s, z), z \sim \mathcal{N}(0, 1) \qquad (12)$$

To achieve this, we train an additional encoder $\mu_\delta(z \mid s)$, which we refer to as latent safety encoder policy conditioned on the state and it learns the distribution of safety embedding such that the expected reward-return is maximized.

$$\mu_\delta^*(z \mid s) = \arg\max_\mu Q_r(s, a), a \sim p_\beta(\cdot \mid s, z), z \sim \mu(\cdot \mid s) \qquad (13)$$

**Advantage Weighted Regression in Latent Space.** To learn the reward-maximizing safety embeddings, we train the encoder $\mu_\delta(z \mid s)$ using reward advantage-weighted regression (AWR). The

optimization objective is given by

$$\mathcal{L}_\mu(\delta) = \mathbb{E}_{(s,a)\sim\mathcal{D}}\left[-\exp\left(\zeta\left(Q_\theta^r(s,a) - V_\phi^r(s)\right)\right)\log p_\beta(a \mid s, z)\right], z \sim \mu_\delta(. \mid s) \qquad (14)$$

Here $\zeta$ is the inverse temperature hyperparameter for the above AWR. While this objective focuses on maximizing the reward-return, it is equally essential to enforce the latent safety constraint to certify policy safety. To ensure that the embeddings produced by the encoder $\mu_\delta$ lie within the high-probability regions of the latent safety space, we employ a $tanh$ squashing function, which restricts the outputs to the range $(-\epsilon, \epsilon), \epsilon > 0$. The resulting policy, which we refer to as LSPC-O, optimizes reward-return within the restricted latent safety constraints space. The LSPC-O policy is parameterized by latent safety encoder policy parameter $\delta$ and CVAE decoder parameter $\beta$. During the gradient step to train the latent safety encoder policy ($\mu_\delta$), the CVAE decoder parameter is kept frozen, but the gradients are allowed to pass through.

An overview of the training process showing each gradient step is provided in Algorithm 1. It is important to note that the policy extraction steps for both the CVAE policy and the latent safety encoder policy do not affect the value function training as defined in section 2.3, and therefore the policy training can be done concurrently with the critics in Actor-Critic framework or after temporal-difference (TD) training of the critics. Detailed implementation specifics can be found in Appendix A.3.

---

**Algorithm 1:** LSPC Training

**Initialize:** $\phi, \theta, \eta, \psi, \alpha, \beta, \delta$
**for** *each gradient step* **do**
    TD learning with IQL:
    $\phi \leftarrow \phi - \nu_\phi \nabla_\phi \mathcal{L}_V(\phi)$
    $\theta \leftarrow \theta - \nu_\theta \nabla_\theta \mathcal{L}_Q(\theta)$
    $\eta \leftarrow \eta - \nu_\eta \nabla_\eta \mathcal{L}_V^c(\eta)$
    $\psi \leftarrow \psi - \nu_\psi \nabla_\psi \mathcal{L}_Q^c(\psi)$
    Policy extraction with AWR:
    $\{\alpha, \beta\} \leftarrow \{\alpha, \beta\} - \nu_{\alpha\beta}\nabla_{\{\alpha,\beta\}}\mathcal{L}_{p,q}(\alpha, \beta)$
    $\delta \leftarrow \delta - \nu_\delta \nabla_\delta \mathcal{L}_\mu(\delta)$

---

## 4 THEORETICAL ANALYSIS

We provide theoretical analysis for LSPC-O as the safe optimal policy is our target, while similar techniques can be used for LSPC-S. In what follows, we begin by imposing assumptions regarding the distances between various policies. These assumptions will allow us to derive performance bounds and sample complexity guarantees for learned policies, ensuring both safety and reward maximization. The metrics for evaluation are based on Eq. 2 such that we adopt $V_r^{\pi^*}(\rho_0) - V_r^\pi(\rho_0)$ and $V_c^\pi(\rho_0) - \kappa$. In light of the definition for the stationary state-action distribution and the fact that $\pi^* = \text{argmax}_\pi \mathbb{E}_{\tau\sim\pi}[\sum_{t=1}^T \gamma^t r(s_t, a_t)]$, we have $\pi^* = \text{argmax}_\pi \mathbb{E}_{(s,a)\sim d^\pi}[r(s,a)]$. This will assist in the analysis presented later. All necessary proof is deferred to the Appendix A.1.

**Assumption 1.** *Suppose that the policy $\pi$ is optimized by Eq. 2 and that the safe policy $\pi_s$ is induced by CVAE. We have $D_{KL}(\pi||\pi_s) \leq \varepsilon_1'$.*

**Assumption 2.** *Suppose that the policy $\pi^*$ is the optimal policy and that the safe policy $\pi_s$ is induced by CVAE. We have $D_{KL}(\pi_s||\pi^*) \leq \varepsilon_2'$.*

In Yao et al. (2024), the authors resorted to similar assumptions involving $\pi_b$ instead of $\pi_s$ (see Appendix A.1.1 for detail) to ensure behavior regularization and establish constraint violation bound. Immediately, we can obtain that the upper bounds for $D_{KL}(\pi||\pi_b)$ and $D_{KL}(\pi_b||\pi^*)$ are looser, since intuitively, $\pi_s$ should be closer to $\pi$ and $\pi^*$ than $\pi_b$. One may argue that when we resort to $\pi_s$ in the above assumptions, whether the behavior regularization can still be satisfied. This is affirmatively true as $\pi_s$ is a safe reconstruction of $\pi_b$ with the generative property of the CVAE. The designed loss also enables the appropriate latent safety embedding construction for training $\pi$. Alternatively, some pre-processing can be done to $\mathcal{D}$ such that only the safe data samples are utilized to learn the policy, as done in BC-Safe Liu et al. (2023a). This may be more conservative to guarantee the safety constraint, but with the compromise of optimality, as an unsafe sample could lead to the higher returns. Therefore, we select the first approach for training in our proposed pipeline. We next present a lemma to show the error propagation.

**Lemma 1.** *Suppose that the stationary state distributions for $\pi$ and $\pi^*$ are defined as $d^\pi(s)$ and $d^{\pi^*}(s)$. The following relationship holds*

$$D_{TV}(d^\pi(s)||d^{\pi^*}(s)) \leq \frac{\gamma}{1-\gamma}\mathbb{E}_{s\sim d^{\pi^*}(s)}[D_{TV}(\pi(\cdot|s)||\pi^*(\cdot|s))], \qquad (15)$$

*where $D_{TV}$ is the total variation distance.*

The proof of Lemma 1 follows similarly from Lemma 2 in Xu et al. (2020). This lemma quantifies the relationship between the stationary state distribution discrepancy w.r.t. the policy distribution discrepancy, which is critical to tighten the performance bound presented in the next.

**Lemma 2.** *Define $V_r^\pi(\rho_0) := \mathbb{E}_{s_0 \sim \rho}[V_r^\pi(s_0)]$ given a policy $\pi$. Then, we have the following:*

$$|V_r^\pi(\rho_0) - V_r^{\pi^*}(\rho_0)| \leq \frac{2R_m}{(1-\gamma)^2} \mathbb{E}_{s \sim d^{\pi^*}(s)}[D_{TV}(\pi(\cdot|s)||\pi^*(\cdot|s))]. \tag{16}$$

Different from existing works Cen et al. (2024); Yao et al. (2024) where the authors bounded $D_{TV}(d^\pi(s)||d^{\pi^*}(s))$ by $D_{TV}(d^{\pi_b}(s)||d^{\pi^*}(s))$, thanks to Lemma 1, Lemma 2 states that the performance gap now stems only from the policy distribution discrepancy between $\pi$ and $\pi^*$ and eventually decays if the number of samples in $\mathcal{D}$ is infinite in our sample complexity analysis. Thus, we next upper bound $\mathbb{E}_{s \sim d^{\pi^*}(s)}[D_{TV}(\pi(\cdot|s)||\pi^*(\cdot|s))]$.

**Lemma 3.** *Let Assumptions 1 and 2 hold. We have the following relationship:*

$$\mathbb{E}_{s \sim d^{\pi^*}(s)}[D_{TV}(\pi(\cdot|s)||\pi^*(\cdot|s))] \leq \sqrt{\varepsilon_1'/2} + \sqrt{\varepsilon_2'/2}. \tag{17}$$

Hence, the first main result to reveal the performance gap between any policy $\pi$ optimized by Eq. 2 and the optimal one $\pi^*$ is as follows.

**Theorem 1.** *Let Assumptions 1 and 2 hold. For the policy $\pi$ optimized by Eq. 2 with the dataset $\mathcal{D}$, the performance gap between $\pi$ and $\pi^*$ can be bounded as*

$$V_r^{\pi^*}(\rho_0) - V_r^\pi(\rho_0) \leq \frac{2R_m}{(1-\gamma)^2}(\sqrt{\varepsilon_1'/2} + \sqrt{\varepsilon_2'/2}). \tag{18}$$

Theorem 1 suggests that the performance gap is attributed to the policy distribution discrepancies. This bound is the worst-case bound irrespective of the number of samples in $\mathcal{D}$. However, based on traditional supervised learning, more data (which requires a weak assumption) should reduce the error such that $\pi$ is closer to $\pi^*$. We will investigate this later and show it is indeed the case. In this context, we analyze the constraint violation bound. Due to Eq. 2, the safety constraint is dictated by the threshold value $\kappa$. Nevertheless, as the policy learning proceeds, such a constraint may or may not be violated, particularly at the early phase of learning. We aim at deriving the similar worst-case bound.

**Theorem 2.** *Let Assumptions 1 and 2 hold. For the policy $\pi$ optimized by Eq. 2 with the dataset $\mathcal{D}$, the constraint violation of $\pi$ can be bounded as*

$$V_c^\pi(\rho_0) - \kappa \leq \frac{2C_m}{(1-\gamma)^2}(\sqrt{\varepsilon_1'/2} + \sqrt{\varepsilon_2'/2}). \tag{19}$$

**Remark 1.** *We notice that both Theorem 1 and Theorem 2 imply that the error bounds for the reward and the cost incurred during learning are w.r.t $\frac{1}{(1-\gamma)^2}(\sqrt{\varepsilon_1'/2} + \sqrt{\varepsilon_2'/2})$, differing in a constant. This intuitively makes sense as we have adopted the same IQL to train both critic networks. The dependence on $\frac{1}{(1-\gamma)^2}$ is necessarily inevitable due to the error propagation from the policy distribution discrepancy to the state distribution discrepancy, which has theoretically been justified in Xu et al. (2020) and Schulman (2015). So far, thanks to Assumption 1 and Assumption 2, the performance gap and the constraint violation are upper bounded. However, would increasing the size of $\mathcal{D}$ reduce them? Particularly, when we have a sufficiently large amount of pre-collected data, can this enforce $V_c^\pi(s) \leq \kappa$ to hold strictly? Inspired by the theory of empirical process Shorack & Wellner (2009), we provide the answers to the above questions and the resulting sample complexity in Appendix A.2.*

**Remark 2.** *Another remark is also provided in this context for the connection between the proposed method and theoretical results. In LSPC, we leverage CVAE to reconstruct safe policies residing in the offline dataset $\mathcal{D}$, strategically ensuring that the further optimal solutions are safe. This justifies Assumption 1 and Assumption 2 that provide new upper bounds for the distribution distances between $\pi$ and $\pi_s$, and $\pi_s$ and $\pi^*$. Previously, instead of $\pi_s$, it was the unknown behavior policy $\pi_b$ Yao et al. (2024), which has less assurance of safety than $\pi_s$ and results in relatively looser bounds. Moreover, in LSPC, we adopt the IQL and AWR for critic model updates and policy extraction, which motivates us to use the value function as the metric to evaluate the proposed algorithm. As the policy is learned via a static offline dataset, the performance gap and constraint violation are assessed based on the stationary distributions, instead of a dynamic metric such as regret Zhong & Zhang (2024). Theorem 1 and Theorem 2 suggest the performance gap and constraint violation during*

*learning are upper bounded by constants that are correlated with the new distribution distance bounds induced by the safety policy $\pi_s$. Additionally, as the policy evaluation and extraction steps are all parametric with deep learning models, the number of data samples plays a central role in controlling the performance, which has been a well-established fact in modern deep learning community. In offline RL, this motivates the need to assess performance gap and constraint violations with respect to the size of $\mathcal{D}$. Thereby, Theorem 3 and Theorem 4 (In Appendix A.2) imply the decay rate with respect to the size of $\mathcal{D}$. Though the analysis can apply to related algorithms with some adaptation, the constants in these results, particularly $\varepsilon_1'$ and $\varepsilon_2'$, are uniquely defined and thus the theoretical conclusions are tailored to the proposed algorithm.*

## 5 RESULTS AND DISCUSSIONS

**Datasets and Metrics.** This section presents a thorough evaluation of our method within the context of safe offline reinforcement learning. Our evaluation uses the DSRL benchmark (Liu et al., 2023a), focusing on normalized return and normalized cost to measure performance. The normalized reward return is given by $R = (R_\pi - R_{min})/(R_{max} - R_{min})$, where $R_\pi$ is the undiscounted total reward accumulated in an episode, and $R_{max}$ and $R_{min}$ are constant for a task and represent the maximum and minimum empirical reward return. Similarly, the normalized cost return is given by $C = C_\pi/\kappa$, where $\kappa > 0$ is the targeted cost threshold. The evaluation spans across tasks from Metadrive (Li et al., 2022), Safety Gymnasium (Ji et al., 2023) and Bullet Safety Gym (Gronauer, 2022). More about these tasks and benchmark can be found in appendix A.4 and Liu et al. (2023a). Following the DSRL constraint variation evaluation, each method is evaluated on each dataset with three distinct target cost thresholds and across three random seeds. Although our method is agnostic to cost threshold variation, we adhere to the same evaluation criteria for consistency. A normalized cost return below 1 indicates adherence to safety requirements, with safety being the primary performance criterion. The results in Table 1 use boldface to represent safe agents with normalized costs smaller than 1 and blue color to highlight safe agent(s) with the highest reward return in the dataset. These results demonstrate how our methods not only adhere to safety constraints but also optimize rewards.

Table 1: Comparison of our methods with baselines across benchmark tasks. **Bold** indicates safety, and **blue** denotes both safety and high performance.

| Method | BC-Safe | | CDT | | CPQ | | FISOR | | LSPC-S | | LSPC-O | |
|---|---|---|---|---|---|---|---|---|---|---|---|---|
| Task | reward ↑ | cost ↓ | reward ↑ | cost ↓ | reward ↑ | cost ↓ | reward ↑ | cost ↓ | reward ↑ | cost ↓ | reward ↑ | cost ↓ |
| Metadrive: | | | | | | | | | | | | |
| EasySparse | **0.11** | **0.21** | **0.17** | **0.23** | -0.06 | 0.07 | **0.38** | **0.15** | **0.62** | **0.06** | **0.71** | **0.46** |
| EasyMean | **0.04** | **0.29** | **0.45** | **0.54** | -0.07 | 0.07 | **0.38** | **0.08** | **0.62** | **0.04** | **0.69** | **0.26** |
| EasyDense | **0.11** | **0.14** | **0.32** | **0.62** | -0.06 | 0.03 | **0.36** | **0.08** | **0.55** | **0.06** | **0.68** | **0.37** |
| MediumSparse | **0.33** | **0.30** | 0.87 | 1.10 | -0.08 | 0.07 | **0.42** | **0.07** | **0.96** | **0.32** | **0.94** | **0.12** |
| MediumMean | **0.31** | **0.21** | **0.45** | **0.75** | -0.05 | 0.05 | **0.39** | **0.02** | **0.85** | **0.43** | **0.94** | **0.11** |
| MediumDense | **0.24** | **0.17** | 0.88 | 2.41 | -0.07 | 0.07 | **0.49** | **0.12** | **0.93** | **0.07** | **0.93** | **0.01** |
| HardSparse | 0.17 | 3.25 | **0.25** | **0.41** | -0.05 | 0.06 | **0.30** | **0.00** | **0.50** | **0.24** | **0.54** | **0.47** |
| HardMean | **0.13** | **0.40** | **0.33** | **0.97** | -0.05 | 0.06 | **0.26** | **0.09** | **0.51** | **0.21** | **0.53** | **0.57** |
| HardDense | **0.15** | **0.22** | **0.08** | **0.21** | -0.04 | 0.08 | **0.30** | **0.10** | **0.47** | **0.08** | **0.50** | **0.23** |
| **Average** | **0.18** | **0.58** | **0.42** | **0.80** | -0.06 | 0.06 | **0.36** | **0.08** | **0.67** | **0.17** | **0.72** | **0.29** |
| Safety Gym: | | | | | | | | | | | | |
| CarButton1 | **0.07** | **0.85** | 0.21 | 1.6 | 0.42 | 9.66 | **-0.02** | **0.04** | -0.02 | 0.14 | -0.01 | 0.11 |
| CarButton2 | -0.01 | 0.63 | 0.13 | 1.58 | 0.37 | 12.51 | **0.01** | **0.09** | -0.09 | 0.21 | -0.12 | 0.39 |
| CarGoal1 | **0.24** | **0.28** | 0.66 | 1.21 | 0.79 | 1.42 | **0.49** | **0.12** | **0.22** | **0.23** | **0.31** | **0.40** |
| CarGoal2 | **0.14** | **0.51** | 0.48 | 1.25 | 0.65 | 3.75 | **0.06** | **0.05** | **0.13** | **0.44** | **0.19** | **0.42** |
| CarPush1 | **0.14** | **0.33** | **0.31** | **0.4** | -0.03 | 0.95 | **0.28** | **0.04** | **0.18** | **0.32** | **0.18** | **0.33** |
| CarPush2 | **0.05** | **0.45** | 0.19 | 1.3 | 0.24 | 4.25 | **0.14** | **0.13** | **0.02** | **0.34** | **0.05** | **0.62** |
| SwimmerVel | 0.51 | 1.07 | **0.66** | **0.96** | 0.13 | 2.66 | **-0.04** | **0.00** | **0.50** | **0.08** | **0.44** | **0.14** |
| HopperVel | **0.36** | **0.67** | **0.63** | **0.61** | 0.14 | 2.11 | **0.17** | **0.32** | **0.26** | **0.39** | **0.69** | **0.00** |
| HalfCheetahVel | **0.88** | **0.54** | **1.0** | **0.01** | 0.29 | 0.74 | **0.89** | **0.00** | **0.79** | **0.01** | **0.97** | **0.10** |
| Walker2dVel | **0.79** | **0.04** | **0.78** | **0.06** | 0.04 | 0.21 | **0.38** | **0.36** | 0.56 | 1.28 | **0.76** | **0.02** |
| AntVel | **0.98** | **0.29** | **0.98** | **0.39** | -1.01 | 0.0 | **0.89** | **0.00** | **0.95** | **0.07** | **0.98** | **0.45** |
| **Average** | **0.38** | **0.51** | **0.55** | **0.85** | 0.19 | 3.48 | **0.30** | **0.11** | **0.32** | **0.32** | **0.40** | **0.27** |
| Bullet Safety Gym: | | | | | | | | | | | | |
| BallRun | 0.27 | 1.46 | 0.39 | 1.16 | 0.22 | 1.27 | **0.18** | **0.00** | **0.08** | **0.00** | **0.14** | **0.00** |
| CarRun | **0.94** | **0.22** | **0.99** | **0.65** | 0.95 | 1.79 | **0.73** | **0.04** | **0.72** | **0.00** | **0.97** | **0.13** |
| DroneRun | 0.28 | 0.74 | **0.63** | **0.79** | 0.33 | 3.52 | **0.30** | **0.16** | **0.54** | **0.00** | **0.57** | **0.00** |
| AntRun | 0.65 | 1.09 | **0.72** | **0.91** | 0.03 | 0.02 | **0.45** | **0.00** | **0.29** | **0.04** | **0.44** | **0.45** |
| BallCircle | **0.52** | **0.65** | 0.77 | 1.07 | 0.64 | 0.76 | **0.34** | **0.00** | **0.27** | **0.28** | **0.47** | **0.01** |
| CarCircle | 0.5 | 0.84 | **0.75** | **0.95** | 0.71 | 0.33 | **0.40** | **0.03** | **0.35** | **0.00** | **0.72** | **0.04** |
| DroneCircle | 0.56 | 0.57 | **0.60** | **0.98** | -0.22 | 1.28 | **0.48** | **0.00** | **0.16** | **0.00** | **0.58** | **0.60** |
| AntCircle | 0.40 | 0.96 | 0.54 | 1.78 | **0.00** | **0.00** | **0.20** | **0.00** | **0.13** | **0.02** | **0.45** | **0.40** |
| **Average** | **0.52** | **0.82** | 0.68 | 1.04 | 0.33 | 1.12 | **0.39** | **0.03** | **0.32** | **0.04** | **0.54** | **0.20** |

**Discussion on Empirical Results.** In the comparative analysis presented in Table 1, our proposed method, LSPC-O, demonstrates a significant performance improvement over existing baselines. Among these, FISOR demonstrates consistency in upholding safety even under strict cost thresholds,

as it imposes hard constraints in the RL problem formulation. However, this approach leads to excessively conservative policies, significantly limiting reward-return performance. Another method CDT conditions the policy on target reward and cost returns. In practice, this method is often unreliable because it is challenging to satisfy both conditions simultaneously. Although the authors suggest improvements, our empirical results show frequent safety violations, especially under smaller cost thresholds. Additionally, interpreting results across different prompt conditions is difficult, making prompt selection a non-trivial task in CDT implementation. BC-Safe applies behavior cloning to a safe subset of the dataset, but it lacks consistent safety and offers no empirical guarantees. Its primary limitations are the need for *sufficient* quantity of 'safe' data and retraining with the filtered data when the cost threshold is changed. Despite being agnostic to the reward labels in the dataset, this method often outperforms FISOR and CDT in performance, as evident in the table 1.

Our method effectively addresses the limitations observed in these baselines. Unlike SafeBC, we employ weighted regression to derive a safe policy, which theoretically only relies on a much weaker prerequisite on the explicit presence of the safe data points (Assumption 6), while also ensuring policy improvement using the reward-labels. Unlike CDT, our method disentangles safety and reward objectives, providing reliable safety guarantees across different tasks and cost thresholds. Additionally, we derive a safe policy in our methodology that can be utilized in safety-critical operations, where constraint satisfaction is prioritized over reward maximization. Finally, compared to FISOR, the conservativeness in our methods can be controlled by adjusting the degree of restriction in the latent safety constraint space, yielding better reward performance while maintaining safety.

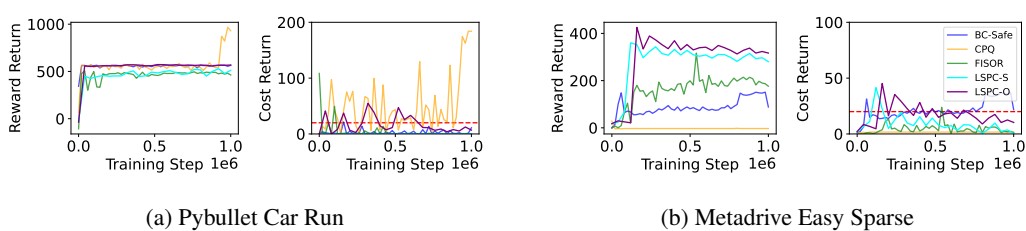

(a) Pybullet Car Run                    (b) Metadrive Easy Sparse

Figure 2: Training progress illustrating how the reward return and cost return evolve over the course of training, highlighting the agent's accumulated reward and adherence to safety constraints.

The graphs in Figure 2 illustrate the evolution of both reward return and cost return as training progresses, comparing how various algorithms perform in adhering to safety constraints while optimizing for reward across two tasks: Pybullet Car Run and Metadrive Easy Sparse. Our method, LSPC-S, demonstrates a conservative approach with a strong emphasis on safety, while LSPC-O achieves a favorable balance between reward maximization and cost minimization, consistently outperforming baseline methods. This is particularly evident in its lower average episode costs as training progresses. In contrast, FISOR fails to outperform even the conservative LSPC-S in both tasks. Behavior Cloning with safe trajectories (BC-Safe) performs reasonably well in the Car Run task, but shows poor performance in Metadrive Easy Sparse, struggling with both reward accumulation and safety adherence. CPQ achieves a higher reward in Car Run, though at the cost of increased episode costs near the end of training. However, in the Metadrive task, CPQ's performance is trivially safe, with near-zero reward accumulation.

Figure 3 illustrates how the learned policies operate in the action space for the Pybullet Car Run and Metadrive Easy Sparse tasks. The plots on the left show kernel density estimates for three policies: (1) the CVAE policy, (2) the safe policy after applying Latent Safety Prioritized Constraints (LSPC), and (3) the optimal policy within the restricted action space defined by LSPC. The right plots display scatter plots of actions sampled from the CVAE policy, convex hull estimates of actions sampled from the safe policy, and the optimal action predicted by LSPC-O, the reward-optimized policy. The color map represents the reward values $Q_r(s, a)$ of the sampled actions for the given state. This representation underscores the ability of our method to prioritize safety while maximizing reward, as the policy concentrates on regions of high $Q_r$ within the restricted latent space.

## 6 RELATED WORKS

Offline RL has demonstrated success in domains such as robotics, healthcare, and recommendation systems (Tarasov et al., 2024; Prudencio et al., 2023; Lange et al., 2012). However, the absence of online interactions can lead to out-of-distribution (OOD) actions that overestimate the value function

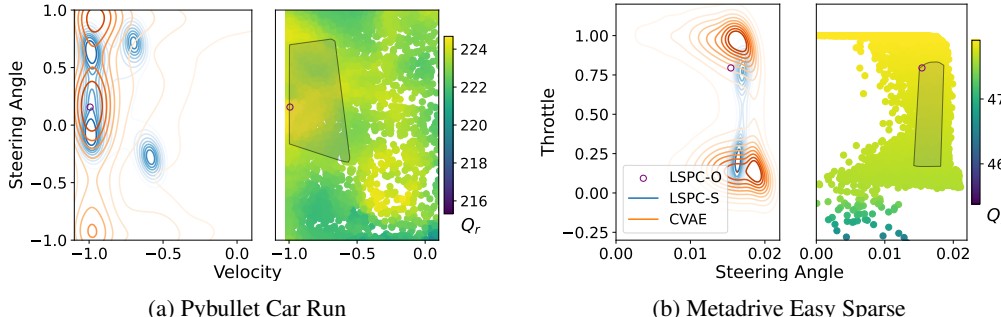

(a) Pybullet Car Run  (b) Metadrive Easy Sparse

Figure 3: Action-space distributions for (a) Pybullet Car Run and (b) Metadrive Easy Sparse. Kernel density estimates (left) and convex hulls within the scatter plot (right) visualize the unconstrained CVAE policy, the constrained safe policy (LSPC-S), and the optimal policy (LSPC-O) identified by the latent safety encoder.

due to unseen data (Levine et al., 2020). Solutions include value function regularization (Kumar et al., 2020; 2019; Kostrikov et al., 2021a), policy constraints to stay near the behavior policy (Fujimoto & Gu, 2021; Wu et al., 2019; Fujimoto et al., 2019), and avoidance of OOD actions' evaluations (Chen et al., 2021; Kostrikov et al., 2021b; Janner et al., 2021). Generative models like CVAE (Fujimoto et al., 2019; Zhou et al., 2021) and diffusion-based methods (Janner et al., 2022; Wang et al., 2022) have also been applied to sample actions in the latent space, reducing OOD occurrences.

Safe Offline RL aims to ensure safety within offline RL by incorporating techniques that balance policy improvement and safety constraints (Liu et al., 2023a). Methods such as COptiDICE (Lee et al., 2022) and OASIS (Yao et al., 2024) extend distribution shaping (Lee et al., 2021) to constrained RL settings. Lagrangian-based approaches like CPQ (Xu et al., 2022a), BCQ-Lag, and BEAR-Lag (Liu et al., 2023a) integrate safe policy learning within offline RL methods like CQL (Kumar et al., 2020), BCQ (Fujimoto et al., 2019) and BEAR (Kumar et al., 2019). Other approaches, such as VOCE (Guan et al., 2024), use probabilistic inference with non-parametric variational distributions (similar to Liu et al. (2022)), while models like CDT (Liu et al., 2023b) and Saformer (Zhang et al., 2023) leverage decision-transformer (Chen et al., 2021) frameworks for cost-prompted sequence modeling. Other generative model-based baselines in safe offline RL include CPQ, which uses a CVAE for behavior policy modeling, and FISOR (Zheng et al., 2024), which employs a diffusion model (Ho et al., 2020) as an actor, trained to select actions based on feasible region identification.

# 7 CONCLUSIONS

In this work, we introduced the Latent Safety-Prioritized Constraints (LSPC) framework for safe offline reinforcement learning, addressing the critical challenge of balancing reward maximization with stringent safety constraints. By leveraging Conditional Variational Autoencoders to model the latent safety constraints, we enable a principled approach to enforce safety while optimizing cumulative rewards within the constraint space. Our method offers a robust solution that avoids the pitfalls of overly restrictive or loose constraints, leading to policies that are both safe and high-performing. Theoretical analysis of policy performance illustrates that the reconstructed safety policy from the unknown behavior policy shrinks the performance gap induced by the value function. The additional sample complexity analysis also reveals the decay rate of the gap w.r.t. the number of samples. Extensive empirical evaluations demonstrate that LSPC-O consistently outperforms existing/recent methods on challenging benchmarks.

Although our proposed methods demonstrate strong theoretical backing and empirical performance, a key limitation may lie in determining how restrictive the latent space for the optimal policy search should be (controlled by $\epsilon$). Future work could focus on deriving theoretical bounds that establish a relationship between this restrictiveness and the cost value, providing both theoretical insights and practical guidelines. From an application perspective, transferring learned policies to real-world robotic tasks via sim2real or few-shot fine-tuning is a promising avenue. Additionally, exploring safety prioritization dynamics in multi-agent cooperative or adversarial settings offers another exciting direction for both theory and practice.

ACKNOWLEDGMENTS

This work was partly supported by the National Science Foundation, USA, under grants NSF CNS-2313104, CAREER CNS-1845969 and CPS Frontier CNS-1954556.

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

# A APPENDIX

## A.1 ADDITIONAL ANALYSIS

In this section, we provide additional analysis and numerical results to support or validate our proposed algorithm. In what follows, we provide the missing proof for the main theoretical results.

### A.1.1 AUXILIARY ASSUMPTIONS

We present the following assumptions for completeness and illustration, though we will not use them for characterizing the main results.

**Assumption 3.** *There exists a constant $\varepsilon_1 > 0$ such that for the policy $\pi$ optimized by Eq. 2 and the behavior policy $\pi_b$, they satisfy the following relationship*

$$D_{KL}(\pi||\pi_b) \leq \varepsilon_1. \tag{20}$$

**Assumption 4.** *There exists a constant $\varepsilon_2 > 0$ such that for the optimal policy $\pi^*$ and the behavior policy $\pi_b$, they satisfy the following relationship*

$$D_{KL}(\pi_b||\pi^*) \leq \varepsilon_2. \tag{21}$$

Assumption 3 retains the behavior regularization such that the distribution shift will not be significant. While Assumption 4 secures a region where $\pi$ is a valid policy. Due to the non-asymptotic manner, $\pi^*$ may not be practically achievable. However, such an optimal policy should not be far away from $\pi_b$. Otherwise, the distribution drift will degrade the model performance. These two assumptions have been used in a recent work Yao et al. (2024) to dictate the performance gap. Due to the intermediate safe policy $\pi_s$ we introduce in this work, the performance gap should be narrowed. Since intuitively, $\pi_s$ should be closer to $\pi$ and $\pi^*$ than $\pi_b$.

### A.1.2 MISSING PROOFS IN SECTION 4

For completeness, we restate all lemmas and theorems in this context.

**Lemma 2** Define $V_r^\pi(\rho_0) := \mathbb{E}_{s_0 \sim \rho}[V_r^\pi(s_0)]$ given a policy $\pi$. Then, we have the following relationship:

$$|V_r^\pi(\rho_0) - V_r^{\pi^*}(\rho_0)| \leq \frac{2R_m}{(1-\gamma)^2} \mathbb{E}_{s \sim d^{\pi^*}(s)}[D_{TV}(\pi(\cdot|s)||\pi^*(\cdot|s))]. \tag{22}$$

*Proof.* In light of the stationary state-action distribution, the following relationship is obtained

$$|V_r^\pi(\rho_0) - V_r^{\pi^*}(\rho_0)| = \frac{1}{1-\gamma}|\mathbb{E}_{(s,a) \sim d^\pi(s,a)}[r(s,a)] - \mathbb{E}_{(s,a) \sim d^{\pi^*}(s,a)}[r(s,a)]|$$

$$\leq \frac{R_m}{1-\gamma} \sum_{(s,a)} |d^\pi(s,a) - d^{\pi^*}(s,a)|$$

$$= \frac{2R_m}{1-\gamma} D_{TV}(d^\pi(s,a)||d^{\pi^*}(s,a))$$

$$\leq \frac{2R_m}{1-\gamma}(D_{TV}(d^\pi(s,a)||d^{\pi^*}(s) \cdot \pi(\cdot|s)) + D_{TV}(d^{\pi^*}(s) \cdot \pi(\cdot|s)||d^{\pi^*}(s,a)))$$

$$= \frac{2R_m}{1-\gamma} D_{TV}(d^\pi(s)||d^{\pi^*}(s)) + \frac{2R_m}{1-\gamma} \mathbb{E}_{s \sim d^{\pi^*}(s)}[D_{TV}(\pi(\cdot|s)||\pi^*(\cdot|s))]$$

$$\leq \frac{2R_m}{(1-\gamma)^2} \mathbb{E}_{s \sim d^{\pi^*}(s)}[D_{TV}(\pi(\cdot|s)||\pi^*(\cdot|s))]. \tag{23}$$

The first inequality is due to the maximum immediate reward. The second inequality follows from the Triangle inequality. The last inequality is based on Lemma 1. This completes the proof. □

**Lemma 3** Let Assumptions 1 and 2 hold. We have the following relationship:

$$\mathbb{E}_{s \sim d^{\pi^*}(s)}[D_{TV}(\pi(\cdot|s)||\pi^*(\cdot|s))] \leq \sqrt{\frac{\varepsilon_1'}{2}} + \sqrt{\frac{\varepsilon_2'}{2}}. \tag{24}$$

*Proof.* Based on the Triangle inequality, the following is attained

$$D_{TV}(\pi(\cdot|s)||\pi^*(\cdot|s)) \leq D_{TV}(\pi(\cdot|s)||\pi_s(\cdot|s)) + D_{TV}(\pi_s(\cdot|s)||\pi^*(\cdot|s)). \tag{25}$$

Thus,

$$
\begin{aligned}
&\mathbb{E}_{s \sim d^{\pi^*}(s)}[D_{TV}(\pi(\cdot|s)||\pi^*(\cdot|s))] \\
&\leq \mathbb{E}_{s \sim d^{\pi^*}(s)}[D_{TV}(\pi(\cdot|s)||\pi_s(\cdot|s))] + \mathbb{E}_{s \sim d^{\pi^*}(s)}[D_{TV}(\pi_s(\cdot|s)||\pi^*(\cdot|s))] \\
&\leq \mathbb{E}_{s \sim d^{\pi^*}(s)}[\sqrt{D_{KL}(\pi(\cdot|s)||\pi_s(\cdot|s))/2}] + \mathbb{E}_{s \sim d^{\pi^*}(s)}[\sqrt{D_{KL}(\pi_s(\cdot|s)||\pi^*(\cdot|s))/2}] \\
&\leq \sqrt{\mathbb{E}_{s \sim d^{\pi^*}(s)}[D_{KL}(\pi(\cdot|s)||\pi_s(\cdot|s))]/2} + \sqrt{\mathbb{E}_{s \sim d^{\pi^*}(s)}[D_{KL}(\pi_s(\cdot|s)||\pi^*(\cdot|s))]/2} \\
&\leq \sqrt{\frac{\varepsilon_1'}{2}} + \sqrt{\frac{\varepsilon_2'}{2}}.
\end{aligned}
\tag{26}
$$

The second inequality follows from the Pinsker's inequality and the third is based on the Jensen's inequality. This completes the proof. $\qquad\square$

**Theorem 1** Let Assumptions 1 and 2 hold. For the policy $\pi$ optimized by Eq. 2 with the dataset $\mathcal{D}$, the performance gap between $\pi$ and $\pi^*$ can be bounded as

$$V_r^{\pi^*}(\rho_0) - V_r^\pi(\rho_0) \leq \frac{2R_m}{(1-\gamma)^2}(\sqrt{\frac{\varepsilon_1'}{2}} + \sqrt{\frac{\varepsilon_2'}{2}}). \tag{27}$$

*Proof.* The conclusion is immediately obtained by combining the conclusions from Lemma 2 and Lemma 3. $\qquad\square$

To derive the constraint violation bound, we first need to establish a similar conclusion as in Lemma 2 based on the cost value functions $V_c^\pi(\rho_0)$ and $V_c^{\pi^*}(\rho_0)$.

**Lemma 4.** *Define $V_c^\pi(\rho_0) := \mathbb{E}_{s_0 \sim \rho_0}[V_c^\pi(s_0)]$ given a policy $\pi$. Then, we have the following relationship:*

$$|V_c^\pi(\rho_0) - V_c^{\pi^*}(\rho_0)| \leq \frac{2C_m}{(1-\gamma)^2}\mathbb{E}_{s \sim d^{\pi^*}(s)}[D_{TV}(\pi(\cdot|s)||\pi^*(\cdot|s))]. \tag{28}$$

*Proof.* The proof follows similarly from that in Lemma 2. $\qquad\square$

**Theorem 2** Let Assumptions 1 and 2 hold. For the policy $\pi$ optimized by Eq. 2 with the dataset $\mathcal{D}$, the constraint violation of $\pi$ can be bounded as

$$V_c^\pi(\rho_0) - \kappa \leq \frac{2C_m}{(1-\gamma)^2}(\sqrt{\frac{\varepsilon_1'}{2}} + \sqrt{\frac{\varepsilon_2'}{2}}). \tag{29}$$

*Proof.* According to Assumption 1 and Assumption 2, combining the conclusion from Lemma 4, we have

$$|V_c^\pi(\rho_0) - V_c^{\pi^*}(\rho_0)| \leq \frac{2C_m}{(1-\gamma)^2}(\sqrt{\frac{\varepsilon_1'}{2}} + \sqrt{\frac{\varepsilon_2'}{2}}) \tag{30}$$

Combining the above inequality with the fact that the optimal policy is constraint satisfactory, i.e., $V_c^{\pi^*}(\rho_0) \leq \kappa$, yields the desirable result. $\qquad\square$

## A.2 ANALYSIS FOR DECAY RATE AND SAMPLE COMPLEXITY

Recalling the following relationship and applying the importance sampling to it, we have:

$$
\begin{aligned}
|V_r^\pi(\rho_0) - V_r^{\pi^*}(\rho_0)| &\leq \frac{2R_m}{(1-\gamma)^2}\mathbb{E}_{s \sim d^{\pi^*}(s)}[D_{TV}(\pi(\cdot|s)||\pi^*(\cdot|s))] \\
&= \frac{2R_m}{(1-\gamma)^2}\mathbb{E}_{s \sim d^{\pi_b}(s)}[\frac{d^{\pi^*}(s)}{d^{\pi_b}(s)}D_{TV}(\pi(\cdot|s)||\pi^*(\cdot|s))] \\
&\leq \frac{2R_m}{(1-\gamma)^2}\mathbb{E}_{s \sim d^{\pi_b}(s)}[D_{TV}(\pi(\cdot|s)||\pi^*(\cdot|s))].
\end{aligned}
\tag{31}
$$

The last inequality follows from $\mathbb{E}_{s \sim d^{\pi_b}(s)}\left[\frac{d^{\pi^*}(s)}{d^{\pi_b}(s)}\right] = 1$. The reason why we use $\pi_b$ instead of $\pi_s$ has two folds. First, the action of $\pi$ is not sampled from actions induced by $\pi_s$ as the extra encoder block in the pipeline also takes input directly from $\mathcal{D}$. Second, directly using $d^{\pi_s}(s)$ will yield a relationship about how the performance gap varies along with the size of a dataset only comprising safe samples. However, this is not practically feasible since $\mathcal{D}$ can contain samples generated by both safe and unsafe policies. Next, we will derive how $\mathbb{E}_{s \sim d^{\pi_b}(s)}[D_{TV}(\pi(\cdot|s)||\pi^*(\cdot|s))]$ evolves with the size of $\mathcal{D}$. With the Pinsker's inequality, $D_{TV}(\cdot||\cdot)$ can be converted to $D_{KL}(\cdot||\cdot)$. Hence, to achieve the best performance $V_r^{\pi^*}(\rho_0)$, it is equivalent to minimizing $\mathbb{E}_{s \sim d^{\pi_b}(s)}[D_{KL}(\pi(\cdot|s)||\pi^*(\cdot|s))]$ by using $\mathcal{D}$. Suppose that the size of $\mathcal{D}$ is $N$ and that the learned policy is parameterized by $w \in \mathbb{R}^n$, such that the following minimization problem can be obtained:

$$\min_{w \in \mathbb{R}^n} \frac{1}{N} \sum_{i=1}^{N} D_{KL}(\pi_w(\cdot|s_i)||\pi^*(\cdot|s_i)). \tag{32}$$

Next, we start with the an assumption for the function of KL-divergence.

**Assumption 5.** *Suppose that the learned policy optimized by Eq. 2 is parameterized by $w \in \mathbb{R}^n$ and that its parameter space is denoted by $\mathcal{W} \subset \mathbb{R}^n$. Let $v_w(s) = D_{KL}(\pi_w(\cdot|s)||\pi^*(\cdot|s))$. The function class $\mathcal{F} := \{v_w(\cdot) : \mathcal{S} \to \mathbb{R}|w \in \mathcal{W}\}$ is a Donsker class such that it satisfies Donsker's theorem Csörgő et al. (2003). Additionally, its variance $\mathbb{V}(v_w(s)|s \sim d^{\pi_b}(s))$ is assumed to be bounded above for all $w \in \mathcal{W}$.*

This assumption is key to obtain the rate of how policy distribution discrepancy between $\pi_w$ and $\pi^*$ decays when the size of $\mathcal{D}$ increases. It is a common assumption when considering training with finite data samples in an empirical process Ma & Kosorok (2005); Geer (2000).

In general, the offline dataset $\mathcal{D}$ should include samples produced by the safe and unsafe policies. However, unfortunately, in many existing works, there is no discussion on how good the data quality is, as it will influence the model learning. Denote by $\mathcal{D}_s$ and $\mathcal{D}_u$ the safe and unsafe data subsets in $\mathcal{D}$ and $N_s$ and $N_u$ the corresponding sizes. We now conduct a thought experiment in this context to motivate the following assumption. In a scenario, if $0 < N_s/N \ll 0.01$, which means the number of safe samples are quite small, one may argue how the agent would learn the optimal policy from such a dataset $\mathcal{D}$, while ensuring the safety constraint. Ideally, since we resort to the advantage weighted regression in our work, in an *asymptotic* manner with infinitely many iterations, i.e., $T \to \infty$ (or a more practical sense, *a sufficiently large number of iterations*), we should attain $\pi^*$ even when $N_s = 1$. Certainly, if $N_s = 0$, then the learning would fail since there is no safe policy to be induced from $\mathcal{D}$, regardless of what $T$ is. Thus, the above discussion essentially results in a more difficult challenge between the time complexity and the sample complexity, which can be a key future work of interest. In this work, we pay only attention to the sample complexity such that the following assumption should be imposed on the dataset $\mathcal{D}$.

**Assumption 6.** *Consider a pre-collected dataset $\mathcal{D}$ such that its safe data subset $\mathcal{D}_s \neq \emptyset$. Denoting by $N_s$ the size of $\mathcal{D}_s$, the ratio $N_s/N > 0$.*

Though in Assumption 6 we need $\mathcal{D}$ to include at least some safe data samples, we don't really require a specific number due to the proposed method. Compared to BCSafe Liu et al. (2023a), which is only trained with safe trajectories that satisfy the constraints, Assumption 6 is much weaker and more practically feasible. Thus, we arrive at the following main result.

**Theorem 3.** *Let Assumptions 5 and 6 hold. Define $V_r^{\pi}(\rho_0) := \mathbb{E}_{s_0 \sim \rho_0}[V_r^{\pi}(s_0)]$ given a policy $\pi$. For the policy $\pi_w$ optimized by Eq. 32 with a dataset $\mathcal{D}$ consisting of $N$ samples, the performance gap satisfies the following relationship*

$$V_r^{\pi^*}(\rho_0) - V_r^{\pi_w}(\rho_0) \leq \mathcal{O}\left(\frac{1}{N^{0.25+\upsilon}}\right), \tag{33}$$

*for any $\upsilon > 0$, when $N \to \infty$.*

*Proof.* By Pinsker's inequality and Jensen's inequality, we have the following relationship

$$\mathbb{E}_{s \sim d^{\pi_b}(s)}[D_{TV}(\pi(\cdot|s)||\pi^*(\cdot|s))] \leq \sqrt{\mathbb{E}_{s \sim d^{\pi_b}(s)}[D_{KL}(\pi_s(\cdot|s)||\pi^*(\cdot|s))]/2}. \tag{34}$$

Based on the traditional supervised learning, it suffices to solve the empirical risk minimization as shown in Eq. 32 so as to minimize $\mathbb{E}_{s \sim d^{\pi_b}(s)}[D_{KL}(\pi_s(\cdot|s)||\pi^*(\cdot|s))]$ in the upper bound. By Assumption 5, the distribution function induced by the state distribution $d^{\pi_b}(s)$ is $\mathbb{E}_{s \sim d^{\pi_b}(s)}[v_w(s)]$.

Similarly, its empirical distribution function is $\frac{1}{N} \sum_{i=1}^{N} v_w(s_i)$. Hence, based on Donsker's theorem, we can acquire

$$\sqrt{N}(\mathbb{E}_{s \sim d^{\pi_b}(s)}[v_w(s)] - \frac{1}{N} \sum_{i=1}^{N} v_w(s_i)) \sim \mathcal{N}(0, \mathbb{V}(v_w(s)|s \sim d^{\pi_b}(s))) \tag{35}$$

which tells us that the centered and scaled version of $\frac{1}{N} \sum_{i=1}^{N} v_w(s_i)$ converges in distribution to a normal distribution with mean 0 and bounded variance $\mathbb{V}(v_w(s)|s \sim d^{\pi_b}(s))$. Immediately, we have

$$\sqrt{N}(\mathbb{E}_{s \sim d^{\pi_b}(s)}[v_w(s)] - \frac{1}{N} \sum_{i=1}^{N} v_w(s_i)) \xrightarrow{N \to \infty} 0, \text{ in probability.} \tag{36}$$

With this in hand, multiplying Eq. 34 by $N^{0.25}$ can be rewritten as

$$
\begin{aligned}
& N^{0.25} \mathbb{E}_{s \sim d^{\pi_b}(s)}[D_{TV}(\pi(\cdot|s)||\pi^*(\cdot|s))] \\
& \leq N^{0.25} \sqrt{\mathbb{E}_{s \sim d^{\pi_b}(s)}[D_{KL}(\pi_s(\cdot|s)||\pi^*(\cdot|s))]/2} \\
& = N^{0.25} \sqrt{\mathbb{E}_{s \sim d^{\pi_b}(s)}[v_w(s)]/2} \\
& = N^{0.25} \sqrt{\frac{1}{2}\left(\mathbb{E}_{s \sim d^{\pi_b}(s)}[v_w(s)] - \frac{1}{N} \sum_{i=1}^{N} v_w(s_i)\right)} \\
& = \sqrt{\frac{1}{2} N^{0.5}\left(\mathbb{E}_{s \sim d^{\pi_b}(s)}[v_w(s)] - \frac{1}{N} \sum_{i=1}^{N} v_w(s_i)\right)} \xrightarrow{N \to \infty} 0, \text{ in probability.}
\end{aligned} \tag{37}
$$

Thus, the above equation implies that $\mathbb{E}_{s \sim d^{\pi_b}(s)}[D_{TV}(\pi(\cdot|s)||\pi^*(\cdot|s))]$ should decay in a rate $\mathcal{O}\left(\frac{1}{N^{0.25+\upsilon}}\right)$ for any $\upsilon > 0$, when $N$ approaches $\infty$, which completes the proof. $\qquad\square$

In Cen et al. (2024), they also arrived at the similar conclusion, but the performance gap is still dictated by the state distribution discrepancy between $d^{\pi_b}(s)$ and $d^{\pi^*}(s)$. This means their performance gap is reduced when the size of $\mathcal{D}$ expands, but cannot reach to 0. Instead, Our work has further reduced it to 0 by converting the state distribution discrepancy to policy distribution discrepancy, given sufficient time. When applying the similar techniques to the cost value function, we have the following result.

**Theorem 4.** *Let Assumptions 5 and 6 hold. Define $V_c^\pi(\rho_0) := \mathbb{E}_{s_0 \sim \rho_0}[V_c^\pi(s_0)]$ given a policy $\pi$. For the policy $\pi_w$ optimized by Eq. 32 with a dataset $\mathcal{D}$ consisting of $N$ samples, the constraint violation bound satisfies the following relationship*

$$V_c^\pi(\rho_0) - \kappa \leq \mathcal{O}\left(\frac{1}{N^{0.25+\upsilon}}\right), \tag{38}$$

*for any $\upsilon > 0$.*

*Proof.* The proof can be obtained by combining the proof from Theorem 2 and Theorem 3. $\qquad\square$

Surprisingly, when $N \to \infty$, $V_c^\pi(\rho_0) \leq \kappa$, which ensures the safety constraint. This delivers us a useful insight that when $N$ is sufficiently large, the cost constraint violation will be significantly small, even neglected.

**Sample Complexity.** If we set an accuracy threshold for both performance gap and constraint violation bound as $\chi$ for any arbitrarily small constant $\chi > 0$, then roughly the size of $\mathcal{D}$ satisfies $N = \mathcal{O}(\frac{1}{\chi^4})$, which suggests that the sample complexity is much larger than that in HasanzadeZonuzy et al. (2021). This is attributed to the offline manner, while in their case, distribution drift is not an issue. Our result also resembles the claim in Theorem 2 from Xu et al. (2022a). We remark on the sample complexity obtained in this context that it is *the worst case* sample complexity as the positive constant $\upsilon$ can be any value. Practically speaking, it falls into the range of $[0, 0.25]$ as suggested by existing works Nguyen-Tang et al. (2021); Li et al. (2024); HasanzadeZonuzy et al. (2021); Shi & Chi (2024). Our analysis on the sample complexity facilitates the theoretical understanding of safe offline reinforcement learning and offers useful insights for more efficient future algorithm design.

### A.3 ALGORITHM AND IMPLEMENTATION DETAILS

In this section, we outline the algorithmic framework and provide implementation details for our method, focusing on the practical aspects. Our methods build upon the CORL (Tarasov et al., 2024) implementation for Implicit Q-Learning (IQL), with the codebase inspired by the OSRL (Liu et al., 2023a) style. The training process involves sampling a batch $\mathcal{B}$ from the replay buffer containing offline data during each gradient step. The expectations described in the methodology are approximated either by computing means over this batch or by using a Monte Carlo approximation.

We employ two Q-value networks, denoted as $Q_{\theta_1}(s,a)$ and $Q_{\theta_2}(s,a)$, to improve training stability and mitigate overestimation of Q-values. The value function $V_\phi(s)$ is trained using the minimum of the two Q-networks, following a Double Q-Learning approach.

The reward value function $V_\phi(s)$ is trained using the following asymmetric L2 loss function:

$$\mathcal{L}_V(\phi) = \frac{1}{|\mathcal{B}|} \sum_{(s,a)\in\mathcal{B}} \left[ L_\xi^2 \left( \min_{i=1,2} Q_{\hat{\theta}_i}(s,a) - V_\phi(s) \right) \right] \tag{39}$$

Both Q-networks are updated simultaneously using the following loss function:

$$\mathcal{L}_Q(\theta_1, \theta_2) = \frac{1}{|\mathcal{B}|} \sum_{i=1,2} \sum_{(s,a,s')\in\mathcal{B}} \left[ (r + \gamma V_\phi(s') - Q_{\theta_i}(s,a))^2 \right] \tag{40}$$

Similarly, we use two q-value networks for cost and use the maximum of this ensemble. The cost critic networks are updated with the following loss functions:

$$\mathcal{L}_V^c(\eta) = \frac{1}{|\mathcal{B}|} \sum_{(s,a)\in\mathcal{B}} \left[ L_\xi^2 \left( V_\eta^c(s) - \max_{i=1,2} Q_{\hat{\psi}_i}^c(s,a) \right) \right] \tag{41}$$

$$\mathcal{L}_Q^c(\psi_1, \psi_2) = \frac{1}{|\mathcal{B}|} \sum_{i=1,2} \sum_{(s,a,s')\in\mathcal{B}} \left[ \left( c + \gamma V_\eta^c(s') - Q_{\psi_i}^c(s,a) \right)^2 \right] \tag{42}$$

The loss function used for the CVAE policy is approximated as:

$$\mathcal{L}_{p,q}(\alpha, \beta) = \frac{-1}{|\mathcal{B}|} \sum_{(s,a)\in\mathcal{B}} w_c(s,a) \left( \log p_\beta(a \mid s, z) - D_{KL} \left[ q_\alpha(z \mid s, a) \parallel p(z \mid s, a) \right] \right), \tag{43}$$

where $z \sim q_\alpha$ is the latent variable and $w_c(s,a) = \exp\left( \lambda \left( V_\phi^c(s) - Q_\theta^c(s,a) \right) \right)$ is the weight given to the behavior cloning loss, common in policy extraction methods like advantage-weighted regression. This method assigns greater weight to the state-action pairs with cost value ($Q_\theta^c(s,a)$) smaller than the cost value associated with that state ($V_\phi^c(s)$), allowing the policy to selectively imitate safer actions from the dataset. To further motivate the CVAE to learn the distribution of safe state-action and reconstruct safe actions, particularly in metadrive, we assign $w_c(s,a) = 0$ for transitions with either cost values greater than a threshold of 0.02. Other hyperparameters used in our methods are discussed in the section that follows.

Finally, we use a reward-advantage weighted regression loss to train the latent safety encoder policy to predict reward-maximizing embeddings in the latent space:

$$\mathcal{L}_\mu(\delta) = \frac{-1}{|\mathcal{B}|} \sum_{(s,a)\in\mathcal{B}} \left[ \exp \left( \zeta \left( Q_\theta^r(s,a) - V_\phi^r(s) \right) \right) \log p_\beta(a \mid s, z) \right], \, z \sim \mu_\delta(. \mid s) \tag{44}$$

---

**Algorithm 2:** Learning LSPC Policies

---

1 **Input:** Dataset $\mathcal{D}$ and hyperparameters including learning rates $\nu$, soft update rate $\mathcal{T}$ and others
2 **Initialize:** Network parameters $\phi, \theta_1, \theta_2, \eta, \psi_1, \psi_2, \alpha, \beta, \delta$
3 **for** *N Gradient Steps* **do**
4 $\quad$ Sample a batch $\mathcal{B}$ of transitions $(s, a, r, c, s')$ from $\mathcal{D}$
5 $\quad$ Use equation 39 to update the parameters of the reward value function:
$\quad\quad \phi \leftarrow \phi - \nu_\phi \nabla_\phi \mathcal{L}_V(\phi)$
6 $\quad$ Use equation 40 to update the parameters of the reward q-value functions:
$\quad\quad \{\theta_1, \theta_2\} \leftarrow \{\theta_1, \theta_2\} - \nu_\theta \nabla_{\{\theta_1, \theta_2\}} \mathcal{L}_Q(\theta_1, \theta_2)$
7 $\quad$ Use equation 41 to update the parameters of the cost value function:
$\quad\quad \eta \leftarrow \eta - \nu_\eta \nabla_\eta \mathcal{L}_V^c(\eta)$
8 $\quad$ Use equation 42 to update the parameters of the cost q-value functions:
$\quad\quad \{\psi_1, \psi_2\} \leftarrow \{\psi_1, \psi_2\} - \nu_\psi \nabla_{\{\psi_1, \psi_2\}} \mathcal{L}_Q^c(\psi_1, \psi_2)$
9 $\quad$ Use equation 43 to update the parameters of the CVAE encoder and decoder:
$\quad\quad \{\alpha, \beta\} \leftarrow \{\alpha, \beta\} - \nu_{\alpha\beta} \nabla_{\{\alpha, \beta\}} \mathcal{L}_{p,q}(\alpha, \beta)$
10 $\quad$ Use equation 44 to update the parameters of the latent safety encoder:
$\quad\quad \delta \leftarrow \delta - \nu_\delta \nabla_\delta \mathcal{L}_\mu(\delta)$
11 $\quad$ Update the parameters of the target q functions:
$\quad\quad \hat{\theta}_1 \leftarrow (1 - \mathcal{T})\hat{\theta}_1 + \mathcal{T}\hat{\theta}_1, \hat{\theta}_2 \leftarrow (1 - \mathcal{T})\hat{\theta}_2 + \mathcal{T}\hat{\theta}_2$
$\quad\quad \hat{\psi}_1 \leftarrow (1 - \mathcal{T})\hat{\psi}_1 + \mathcal{T}\hat{\psi}_1, \mathcal{T}\psi_2 \leftarrow (1 - \mathcal{T})\hat{\psi}_2 + \mathcal{T}\hat{\psi}_2$
12 **Return:** Trained network parameters $\phi, \theta_1, \theta_2, \eta, \psi_1, \psi_2, \alpha, \beta, \delta$

---

### A.3.1 HYPERPARAMETERS AND TUNING

To facilitate understanding and analysis of hyperparameter effects, we classify them into three broad categories: IQL critic learning hyperparameters, AWR policy extraction hyperparameters, and those specific to our method, LSPC. The IQL and AWR hyperparameters are kept consistent across all experiments in this study. These common hyperparameters are listed in Table 2.

Table 2: Common Hyperparameters for IQL and AWR

| Hyperparameter | Value |
|---|---|
| Batch size ($|\mathcal{B}|$) | 1024 |
| Discount factor ($\gamma$) | 0.99 |
| Soft update rate for Q-networks ($\mathcal{T}$) | 0.005 |
| Inverse temperature for reward | 2.0 |
| Inverse temperature for cost | 2.0 |
| Learning rates for all parameters | $3 \times 10^{-4}$ |
| Asymmetric L2 loss coefficient ($\xi$) | 0.7 |
| Max exp advantage weight (both cost and reward) | 200.0 |

For the LSPC-specific hyperparameters, the values are chosen to balance the performance and safety constraints while trying to minimize the need for extensive tuning. The KL divergence coefficient in VAE loss functions regularizes the latent space by encouraging the learned distribution to approximate a prior distribution, balancing reconstruction accuracy with structured latent representations for effective data generation. We set this to 0.5 in all of our experiments. The dimension of the latent safety space is fixed to 32, the general idea being that the space should be of sufficient capacity to capture the safety-wise distribution of the state-action in the dataset. The degree of restriction that constrains the policy to operate within the high-likelihood region of the latent safety space is set to 0.25 for LSPC-S. Generally, the same restriction value of 0.25 works for LSPC-O as well. However, for BulletSafety environments (except AntRun), this restriction is relaxed to 0.6 to enhance performance against baselines, given the upperhand our method has in managing cost returns. While we strive for consistent hyperparameters across tasks, the diversity of the benchmark tasks, with each using different simulators, dynamics, and objectives, might require some adjustments.

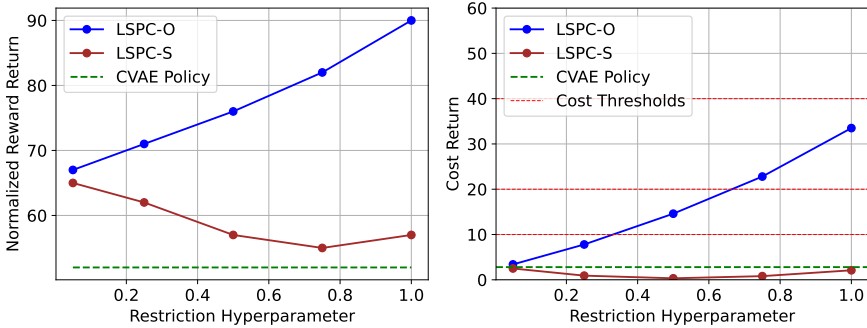

Figure 4: Effect of variation of the latent space restriction hyperparameter while evaluating in Metadrive Easy Sparse environment. The red dashed horizontal lines represent the cost thresholds suggested by Liu et al. (2023a) for this environment.

Figure 4 illustrates the impact of varying the restriction hyperparameter for both LSPC-S and LSPC-O. As the latent safety restriction is increased, the reward performance of LSPC-O improves progressively relative to LSPC-S and CVAE Policy. This is because the latent safety encoder policy in LSPC-O can explore a larger region within the latent space to find reward-maximizing actions. However, this improved performance comes with a trade-off, i.e., higher cost returns. The figure demonstrates how different settings of the restriction hyperparameter yield different LSPC-O policies, each safe under certain cost thresholds used in our experiments.

### A.3.2 ABLATION

Ablation studies are critical for evaluating the influence of individual components of a model on its overall performance. While the main text primarily focuses on presenting the core contributions of our Latent Safety-Constrained Policy (LSPC) framework, this section provides a detailed exploration of specific ablation studies to address the impact of hyperparameters, architectural design choices, and loss functions on performance and safety adherence.

**Role Reversal of CVAE and Safety Encoder.** In this experiment, we investigated the impact of reversing the roles of the CVAE ($\alpha, \beta$) and safety encoder ($\delta$) in the LSPC framework. Specifically, in this configuration (referred to as *Converse LSPC-O*), the CVAE is trained to maximize rewards while the encoder minimizes costs within the inferred latent space. The architecture is detailed in figure 5a, and the training is conducted for 300k timesteps in the HalfCheetah-velocity environment with varying levels of restriction ($\epsilon$) in the latent space. The respective training curves are in figure 5b.

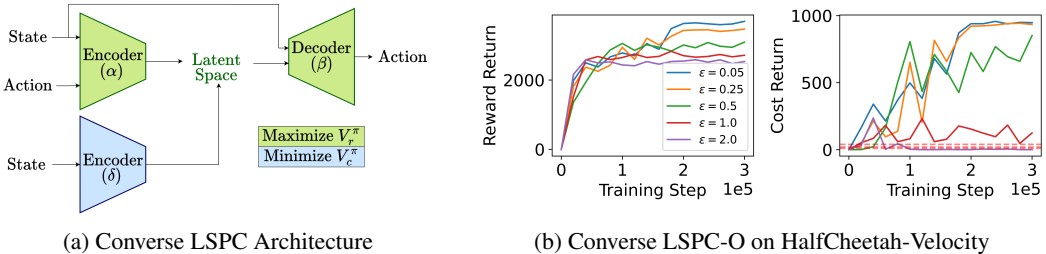

(a) Converse LSPC Architecture      (b) Converse LSPC-O on HalfCheetah-Velocity

Figure 5: Ablation study on role reversal of CVAE and Safety Encoder

With low $\epsilon$ (high restriction), the encoder ($\delta$) struggled to learn an effective cost-reducing policy. As $\epsilon$ increased, some decrease in costs was achieved; however, this came at the expense of lower reward returns. While we can further loosen the restriction to get safer policies, very loose restrictions may cause out-of-distribution (OOD) action generation issues. A general concern in CVAE frameworks is that utilizing latent space samples too far from the mean might cause the decoder to generate out-of-distribution (OOD) samples, as the density of training samples decreases further from the

mean. This highlights the necessity of designing a latent space representation that satisfies both the in-distribution requirement and safety constraints, as addressed in the original LSPC framework. By carefully structuring the roles of the CVAE and safety encoder, the original framework ensures that the inferred latent space boundaries contain safe and in-distribution samples. Moreover, the original framework always maintains a conservative policy (LSPC-S) learned during the training of LSPC-O as a backup. This ablation study underscores the importance of these architectural design choices in balancing performance and safety.

**Inverse Temperature Hyperparameter in LSPC-S.**  In AWR, the inverse temperature hyperparameter $\lambda$ dictates the trade-off between behavior regularization and value function optimization. For LSPC-S, a lower $\lambda$ indicates a higher degree of behavior cloning, while a larger $\lambda$ places sharper weight on samples with better cost advantages. To evaluate the impact of this parameter on both reward performance and safety adherence, we trained LSPC-S with $\lambda$ values of 1, 2, and 4 in the CarRun and BallCircle environments.

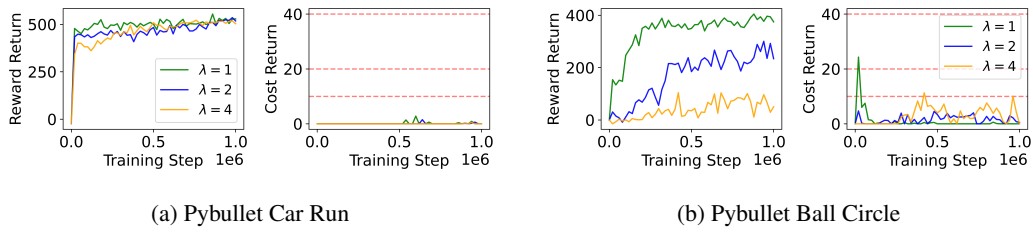

(a) Pybullet Car Run                            (b) Pybullet Ball Circle

Figure 6: Ablation study on the effect of inverse temperature hyperparameter ($\lambda$) in LSPC-S

In the training logs presented in figure 6, it can be observed that safety adherence is consistently maintained by LSPC-S across all tested $\lambda$ values. However, higher $\lambda$ values typically results in restricted reward-wise performance. This effect is particularly pronounced in the BallCircle environment, where lower $\lambda$ values, such as $\lambda = 1$, enable LSPC-S to achieve significantly higher reward returns compared to $\lambda = 2$ and $\lambda = 4$.

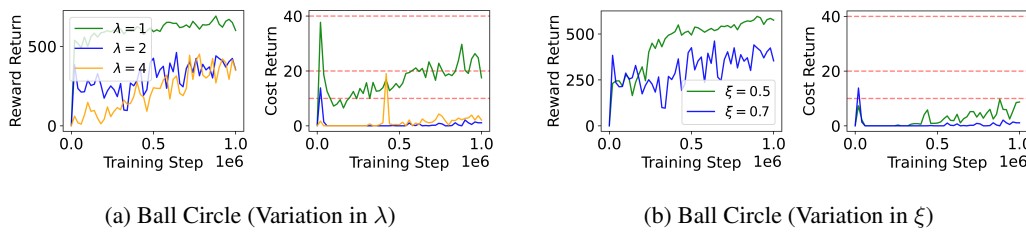

(a) Ball Circle (Variation in $\lambda$)                            (b) Ball Circle (Variation in $\xi$)

Figure 7: Ablation study on how the effect of variation in $\lambda$ and $\xi$ translate to LSPC-O

These findings raise an important question about how the varying reward returns and consistent safety adherence in LSPC-S translate to LSPC-O. Our results indicate that when the CVAE (or LSPC-S) is trained with smaller $\lambda$ values, the reward performance of LSPC-O improves. However, this improvement comes with an increased cost returns, potentially leading to unsafe policies as depicted in figure 7a. While LSPC-S with lower $\lambda$ effectively balances reward and cost returns, the encoder in LSPC-O, trained to further maximize rewards, can exacerbate safety risks. This highlights the critical trade-off between performance and safety optimization when selecting $\lambda$ in the LSPC framework.

**Asymmetric Loss in IQL.**  In this experiment, we analyze the dependence of our method on the asymmetric loss function of IQL, particularly for cost management and safety adherence. As listed in A.3.1, we use a coefficient of $\xi = 0.7$ in the asymmetric loss function used for expectile regression in Implicit Q-Learning (IQL) on our LSPC framework. In LSPC-S, this is intended to discourage the underestimation of cost Q-values. Here, we compare the results against $\xi = 0.5$, which leads to a symmetric Mean Squared Error (MSE) loss, equivalent to SARSA-style policy evaluation.

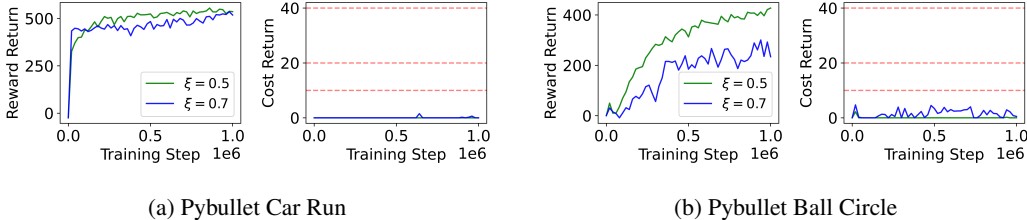

(a) Pybullet Car Run                (b) Pybullet Ball Circle

Figure 8: Ablation study on the effect of IQL expectile loss coefficient ($\xi$) in LSPC-S

As can be observed in figure 8, using the symmetric loss in LSPC-S led to improved reward-wise performance compared to the asymmetric loss (conservative cost estimation), with both $\xi$ values resulting in policies that adhere to the safety requirements. However, for LSPC-O, where an additional encoder is trained to optimize rewards further, the symmetric loss resulted in higher cost returns, indicating chances of safety compromises. These findings depicted in figure 7b highlight a critical trade-off, similar to the ablation with $\lambda$: while less conservative policies with symmetric loss can achieve better rewards, they may translate to increased safety risks in LSPC-O.

### A.3.3   SAFETY AND PERFORMANCE TRADE-OFFS IN LSPC-S AND LSPC-O

The CVAE in the LSPC framework is trained using a cost-advantage weighted ELBO loss, aiming to minimize the cost value function $V_c^\pi$. By sampling actions from high-probability regions in the CVAE latent space, LSPC-S rerpesents a conservative policy that effectively mitigates the risk of out-of-distribution (OOD) actions and generates safe actions. While LSPC-O similarly samples from this restricted latent space and is theoretically expected to maintain equivalent safety, it may introduce some trade-offs in safety by optimizing for reward-maximizing latent variables rather than random latent samples. These trade-offs in safety can often be attributed to the latent safety encoder in LSPC-O exploiting inaccuracies in the action mappings (from the CVAE decoder) to maximize rewards, potentially leading to OOD or unsafe actions. For example, as seen in the ablation study, while LSPC-S ensures safety even under less restrictive hyperparameters, LSPC-O, due to its reward-seeking objective, can exhibit a higher cost-return, often leading to disproportionately higher costs.

Despite these trade-offs, the flexibility of our framework allows for the adjustment of safety requirements via a tunable restriction hyperparameter. A higher restriction (lower $\epsilon$) ensures that samples are drawn from regions closer to the mean of the latent distribution prior (which means stricter LSPC constraint). In addition, when $\epsilon$ is smaller, the encoder in LSPC-O is also forced to operate within a smaller region, resulting in a more restrictive policy. This flexibility in tuning $\epsilon$ according to the safety requirements allows the framework to adapt to different safety conditions for a given task, while still optimizing for rewards within those constraints, whether loose or stringent.

### A.3.4   TRAINING TIME OF THE EXPERIMENTS

The device used for reporting the training times in this section is a Dell Alienware Aurora R12 system with an 11th gen Intel Core i7 processor, 32 GB DDR4, and an NVIDIA GeForce RTX 3070 8GB GPU. All experiments were run on a CUDA device.

LSPC training requires approximately 3.5 hours for 1 million training steps, achieving a rate of around 90 training iterations per second. It is important to note that this reported training time incorporates the training of both LSPC-S and LSPC-O components of our framework. This training time performance is comparable to CPQ under similar conditions and training steps. Behavior Cloning (BC), being more straightforward, is faster, requiring about 35 minutes for 1 million timesteps at 490 training steps per second. CDT training for 100k timesteps takes approximately 3.5 hours, running at about 8 iterations per second.

### A.3.5 Comparison with CPQ

CPQ (Xu et al., 2022a) also employs a CVAE and is referenced multiple times in this paper. Here, we highlight the key differences between our proposed method and CPQ across the following dimensions.

**Value Learning.** CPQ utilizes a cost-sensitive version of CQL (Kumar et al., 2020), which maximizes the cost value for out-of-distribution (OOD) actions while using only constraint-safe and in-distribution-safe samples to learn the reward Q-function. In contrast, our method does not use the constraint-penalized Bellman operator like CPQ. Instead, we adopt IQL for temporal-difference (TD) learning of both reward and cost values, employing standard Bellman updates to ensure simplicity and stability.

**Policy Extraction.** CPQ applies the off-policy deterministic policy gradient method for policy extraction. Our approach, however, employs Advantage Weighted Regression (AWR), where the actor is updated via weighted behavior cloning. This ensures alignment of the policy with the behavior distribution, promoting stability in offline RL settings. Specifically, we train cost-AWR to optimize the CVAE policy and reward-AWR to optimize an additional encoder.

**CVAE Usage.** While both CPQ and our framework incorporate a CVAE, their purposes diverge significantly. CPQ uses the CVAE to detect and penalize OOD actions as high-cost during Q-learning. However, this approach can distort the value function, thereby hindering generalizability and safety adherence, as observed in our experiments and noted in prior works such as DSRL (Liu et al., 2023a) and FISOR (Zheng et al., 2024). By contrast, our method employs the CVAE as a policy model that expressively represents safety constraints in the latent space, rather than as an OOD detection mechanism.

### A.4 Benchmark Details and Additional Baselines

For empirical evaluation of our proposed methods, we use the standard Datasets for Safe Reinforcement Learning (DSRL) benchmarking suite [1] introduced by Liu et al. (2023a). The authors present a comprehensive set of datasets designed to facilitate the development and evaluation of offline safe reinforcement learning algorithms across standard safe RL tasks. This suite includes D4RL-styled (Fu et al., 2020) datasets and robust baseline implementations for offline safe RL, aiding in both training and deployment phases. For baselines, we use the official implementation [2] and hyperparameters for FISOR, as provided by the authors, along with their published results (Zheng et al., 2024). For all other baselines, we rely on the OSRL's official implementation, hyperparameters, and results as reported in the corresponding whitepaper Liu et al. (2023a). OSRL implementations as well as hyperparameter configuration can be found here [3].

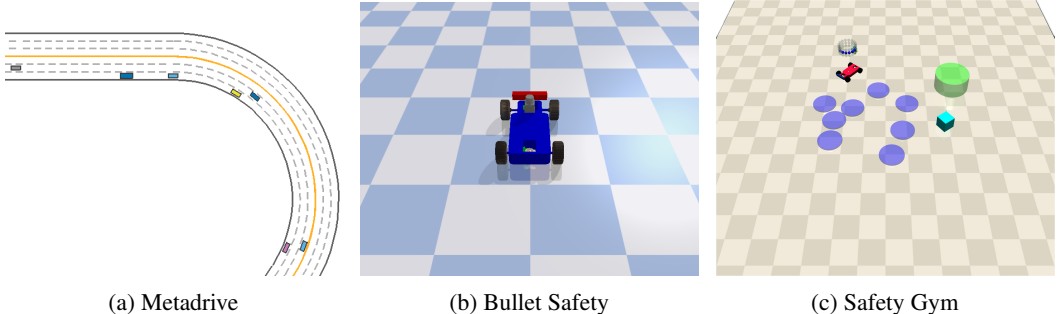

| (a) Metadrive | (b) Bullet Safety | (c) Safety Gym |

Figure 9: Example images from the simulation environments used in the experiments

For evaluation and comparison against the baselines, we use the metrics suggested by the DSRL whitepaper. We use the normalized reward return $R_{\text{normalized}}$, calculated as:

$$R_{\text{normalized}} = \frac{R_\pi - r_{\min}(M)}{r_{\max}(M) - r_{\min}(M)},$$

---

[1] https://github.com/liuzuxin/DSRL
[2] https://github.com/ZhengYinan-AIR/FISOR
[3] https://github.com/liuzuxin/OSRL

where $r_{\max}(M)$ and $r_{\min}(M)$ are the task-specific maximum and minimum empirical returns, respectively. Additionally, we use the normalized cost return $C_{\text{normalized}}$, defined as:

$$C_{\text{normalized}} = \frac{C_\pi + \varsigma}{\kappa + \varsigma},$$

where $\kappa$ is the target cost threshold and $\varsigma$ is a small constant to avoid division by zero. These metrics ensure consistent comparisons across safe RL tasks. Following the DSRL constraint variation evaluation, each method is evaluated on each dataset with three distinct target cost thresholds and across three random seeds. DSRL evaluation uses the average of the normalized reward and cost to better characterize performance under varying conditions; this is particularly relevant for the Lagrange-based algorithms that depend on the cost threshold. The cost threshold values used in DSRL evaluation are set at $[10, 20, 40]$ for MetaDrive and Bullet Safety Gym environments, and $[20, 40, 80]$ for Safety Gymnasium environments. The same criteria are applied in our evaluation tables the normalized cost threshold is set to 1. Bold letters in the evaluation table represent safe agents whose normalized cost is smaller than 1, while blue and bold indicate the safe agent with the highest reward. Additionally, if a safer agent is present within a -0.05 normalized return, that agent is also highlighted in blue and bold.

### A.4.1 METADRIVE

MetaDrive [4] is a self-driving simulator developed using the Panda3D engine, designed to simulate realistic driving environments with complex road structures and dynamic traffic conditions. The autonomous driving tasks in MetaDrive evaluate safety through three primary criteria: (i) collisions, (ii) off-road driving, and (iii) speeding violations. There are nine distinct offline learning tasks, each named in the format {Road}{Vehicle}. The Road category features three difficulty levels for self-driving cars: easy, medium, and hard. The Vehicle category includes four levels of surrounding traffic: sparse, medium, and dense. The stochastic nature of the environment is introduced through the random initialization of traffic patterns and map layouts, providing a challenging and varied benchmark for assessing the performance and safety of reinforcement learning algorithms in autonomous driving scenarios (Li et al., 2022; Liu et al., 2023a). Table 6 summarizes the different configuration and map types used in the different evaluation environments that we use within the MetaDrive self-driving simulator.

Table 3: MetaDrive Performance Metrics

| Method | BC | | BC-Safe | | CDT | | BCQ-Lag | | BEAR-Lag | | CPQ | | COptiDICE | | FISOR | | LSPC-S | | LSPC-O | |
| Task | reward ↑ | cost ↓ | reward ↑ | cost ↓ | reward ↑ | cost ↓ | reward ↑ | cost ↓ | reward ↑ | cost ↓ | reward ↑ | cost ↓ | reward ↑ | cost ↓ | reward ↑ | cost ↓ | reward ↑ | cost ↓ | reward ↑ | cost ↓ |
| easysparse | 0.17 | 1.54 | **0.11** | **0.21** | 0.17 | 0.23 | 0.78 | 5.01 | **0.11** | **0.86** | -0.06 | 0.07 | 0.96 | 5.44 | **0.38** | **0.15** | 0.62 | 0.06 | **0.71** | **0.46** |
| easymean | 0.43 | 2.82 | **0.04** | **0.29** | 0.45 | 0.54 | 0.71 | 3.44 | 0.08 | 0.86 | -0.07 | 0.07 | 0.66 | 3.97 | **0.38** | **0.08** | 0.62 | 0.04 | **0.69** | **0.26** |
| easydense | 0.27 | 1.94 | **0.11** | **0.14** | 0.32 | 0.62 | 0.26 | 0.47 | 0.02 | 0.41 | -0.06 | 0.03 | 0.5 | 2.54 | **0.36** | **0.08** | 0.55 | 0.06 | **0.68** | **0.37** |
| mediumsparse | 0.83 | 3.34 | **0.33** | **0.30** | 0.87 | 1.10 | 0.44 | 1.16 | -0.03 | 0.17 | -0.08 | 0.07 | 0.71 | 2.41 | **0.42** | **0.07** | **0.96** | **0.32** | **0.94** | **0.12** |
| mediummean | 0.77 | 2.53 | **0.31** | **0.21** | 0.45 | 0.75 | 0.78 | 1.53 | 0.00 | 0.34 | -0.08 | 0.05 | 0.76 | 2.05 | **0.39** | **0.02** | **0.85** | **0.43** | **0.94** | **0.11** |
| mediumdense | 0.45 | 1.47 | **0.24** | **0.17** | 0.88 | 2.41 | 0.58 | 1.89 | 0.01 | 0.28 | -0.07 | 0.07 | 0.69 | 2.24 | **0.49** | **0.12** | **0.93** | **0.07** | **0.93** | **0.01** |
| hardsparse | 0.42 | 1.80 | **0.17** | 3.25 | **0.25** | **0.41** | 0.50 | 1.02 | 0.01 | 0.16 | -0.05 | 0.06 | 0.37 | 2.05 | **0.30** | **0.00** | **0.50** | **0.24** | **0.54** | **0.47** |
| hardmean | 0.20 | 1.77 | **0.13** | **0.40** | 0.33 | 0.97 | 0.47 | 2.56 | 0.00 | 0.21 | -0.05 | 0.06 | 0.32 | 2.47 | **0.26** | **0.09** | **0.51** | **0.21** | **0.53** | **0.57** |
| harddense | 0.20 | 1.33 | **0.15** | **0.22** | 0.08 | 0.21 | 0.35 | 1.40 | 0.02 | 0.26 | -0.04 | 0.08 | 0.24 | 1.68 | **0.30** | **0.10** | **0.47** | **0.08** | **0.50** | **0.23** |
| **Average** | 0.42 | 2.06 | **0.18** | **0.58** | 0.42 | 0.80 | 0.54 | 2.05 | 0.02 | 0.39 | -0.06 | 0.06 | 0.58 | 2.77 | **0.36** | **0.08** | **0.67** | **0.17** | **0.72** | **0.29** |

### A.4.2 SAFETY GYMNASIUM

Safety Gymnasium [5] is a collection of environments based on the MuJoCo physics simulator, designed to offer a variety of tasks with adjustable safety constraints and challenges, enabling different levels of difficulty (Ji et al., 2023). We assess the car agent on Button, Goal, and Push tasks, where the numbered suffixes (1 and 2) indicate the difficulty levels. Additionally, Safety Gymnasium includes a subset of Safe Velocity tasks that impose velocity constraints on agents, extending the standard GymMuJoCo locomotion tasks to incorporate safety considerations. We evaluate the methods on Swimmer, Hopper, HalfCheetah, Walker2D, and Ant for the velocity tasks.

### A.4.3 BULLET SAFETY GYM

Bullet Safety Gym [6] is a suite of environments built on the PyBullet physics simulator, designed similarly to Safety Gymnasium but with shorter time horizons and more agents. Unlike Safety

---

[4] https://github.com/metadriverse/metadrive
[5] https://github.com/PKU-Alignment/safety-gymnasium
[6] https://github.com/SvenGronauer/Bullet-Safety-Gym

Table 4: Safety Gym Performance Metrics

| Method | BC | | BC-Safe | | CDT | | BCQ-Lag | | BEAR-Lag | | CPQ | | COptiDICE | | FISOR | | LSPC-S | | LSPC-O | |
|---|---|---|---|---|---|---|---|---|---|---|---|---|---|---|---|---|---|---|---|---|
| Task | reward ↑ | cost ↓ | reward ↑ | cost ↓ | reward ↑ | cost ↓ | reward ↑ | cost ↓ | reward ↑ | cost ↓ | reward ↑ | cost ↓ | reward ↑ | cost ↓ | reward ↑ | cost ↓ | reward ↑ | cost ↓ | reward ↑ | cost ↓ |
| CarButton1 | 0.03 | 1.38 | **0.07** | **0.85** | 0.21 | 1.6 | 0.04 | 1.63 | 0.18 | 2.72 | 0.42 | 9.66 | -0.08 | 1.68 | **-0.02** | **0.04** | **-0.02** | **0.14** | -0.01 | 0.11 |
| CarButton2 | -0.13 | 1.24 | **-0.01** | **0.63** | 0.13 | 1.58 | 0.06 | 2.13 | -0.01 | 2.29 | 0.37 | 12.51 | -0.07 | 1.59 | **0.01** | **0.09** | -0.09 | 0.21 | -0.12 | 0.39 |
| CarGoal1 | **0.39** | **0.33** | 0.24 | 0.28 | 0.66 | 1.21 | **0.47** | **0.78** | 0.61 | 1.13 | 0.79 | 1.42 | 0.35 | 0.54 | **0.49** | **0.12** | 0.22 | 0.23 | 0.31 | 0.40 |
| CarGoal2 | 0.23 | 1.05 | 0.14 | 0.51 | 0.48 | 1.25 | 0.3 | 1.44 | 0.28 | 1.01 | 0.65 | 3.75 | **0.25** | **0.91** | 0.06 | 0.05 | 0.13 | 0.44 | 0.19 | 0.42 |
| CarPush1 | **0.22** | **0.36** | 0.14 | 0.33 | **0.31** | **0.4** | 0.23 | 0.43 | 0.21 | 0.54 | -0.03 | 0.95 | 0.23 | 0.5 | **0.28** | **0.04** | 0.18 | 0.32 | 0.18 | 0.33 |
| CarPush2 | **0.14** | **0.9** | 0.05 | 0.45 | 0.19 | 1.3 | 0.15 | 1.38 | 0.1 | 1.2 | 0.24 | 4.25 | 0.09 | 1.07 | 0.14 | 0.13 | 0.02 | 0.34 | 0.05 | 0.62 |
| SwimmerVel | 0.49 | 4.72 | 0.51 | 1.07 | **0.66** | **0.96** | 0.48 | 6.58 | 0.3 | 2.33 | 0.13 | 2.66 | 0.63 | 7.58 | -0.04 | 0.00 | 0.50 | 0.08 | 0.44 | 0.14 |
| HopperVel | 0.65 | 6.39 | **0.36** | **0.67** | 0.63 | 0.61 | 0.78 | 5.02 | 0.34 | 5.86 | 0.14 | 2.11 | 0.13 | 1.51 | **0.17** | **0.32** | 0.26 | 0.39 | **0.69** | **0.00** |
| HalfCheetahVel | 0.97 | 13.1 | **0.88** | **0.54** | **1.0** | **0.01** | 1.05 | 18.21 | 0.98 | 6.58 | 0.29 | 0.74 | 0.65 | 0.00 | **0.89** | **0.00** | 0.79 | 0.01 | **0.97** | **0.10** |
| Walker2dVel | 0.79 | 3.88 | **0.79** | **0.04** | **0.78** | **0.06** | **0.79** | **0.17** | 0.86 | 3.1 | 0.04 | 0.21 | 0.12 | 0.74 | 0.38 | 0.36 | 0.56 | 1.28 | **0.76** | **0.02** |
| AntVel | 0.98 | 3.72 | **0.98** | **0.29** | **0.98** | **0.39** | 1.02 | 4.15 | -1.01 | 0.0 | -1.01 | 0.0 | 1.0 | 3.28 | **0.89** | **0.00** | **0.95** | **0.07** | **0.98** | **0.45** |
| **Average** | 0.43 | 3.37 | **0.38** | **0.51** | **0.55** | **0.85** | 0.49 | 3.81 | 0.26 | 2.43 | 0.19 | 3.48 | 0.30 | 1.76 | **0.30** | **0.11** | 0.32 | 0.32 | **0.40** | **0.27** |

Gymnasium, which features longer time horizons with shorter simulation steps, Bullet Safety Gym employs shorter time horizons, which may facilitate faster training (Gronauer, 2022). The suite includes four distinct agent types: Ball, Car, Drone, and Ant, as well as two task types: Circle and Run. The tasks are named in the format {Agent}{Task}.

Table 5: Bullet Safety Gym Performance Metrics

| Method | BC | | BC-Safe | | CDT | | BCQ-Lag | | BEAR-Lag | | CPQ | | COptiDICE | | FISOR | | LSPC-S | | LSPC-O | |
|---|---|---|---|---|---|---|---|---|---|---|---|---|---|---|---|---|---|---|---|---|
| Task | reward ↑ | cost ↓ | reward ↑ | cost ↓ | reward ↑ | cost ↓ | reward ↑ | cost ↓ | reward ↑ | cost ↓ | reward ↑ | cost ↓ | reward ↑ | cost ↓ | reward ↑ | cost ↓ | reward ↑ | cost ↓ | reward ↑ | cost ↓ |
| BallRun | 0.6 | 5.08 | 0.27 | 1.46 | 0.39 | 1.16 | 0.76 | 3.91 | -0.47 | 5.03 | 0.22 | 1.27 | 0.59 | 3.52 | **0.18** | **0.00** | 0.08 | 0.00 | **0.14** | **0.00** |
| CarRun | **0.97** | **0.33** | **0.94** | **0.22** | 0.99 | 0.65 | **0.94** | **0.15** | 0.68 | 7.78 | 0.95 | 1.79 | 0.87 | 0.0 | 0.73 | 0.04 | 0.72 | 0.00 | **0.97** | **0.13** |
| DroneRun | 0.24 | 2.13 | **0.28** | **0.74** | **0.63** | **0.79** | 0.72 | 5.54 | 0.42 | 2.47 | 0.33 | 3.52 | 0.67 | 4.15 | **0.30** | **0.16** | 0.54 | 0.00 | 0.57 | 0.00 |
| AntRun | 0.72 | 2.93 | 0.65 | 1.09 | **0.72** | **0.91** | 0.76 | 5.11 | 0.15 | 0.73 | 0.03 | 0.02 | 0.61 | 0.94 | 0.45 | 0.00 | 0.29 | 0.04 | 0.44 | 0.45 |
| BallCircle | 0.74 | 4.71 | **0.52** | **0.65** | 0.77 | 1.07 | 0.69 | 2.36 | 0.86 | 3.09 | 0.64 | 0.76 | 0.70 | 2.61 | 0.34 | 0.00 | 0.27 | 0.28 | **0.47** | **0.01** |
| CarCircle | 0.58 | 3.74 | 0.5 | 0.84 | 0.75 | 0.95 | 0.63 | 1.89 | 0.74 | 2.18 | 0.71 | 0.33 | 0.49 | 3.14 | 0.40 | 0.03 | 0.35 | 0.00 | **0.72** | **0.04** |
| DroneCircle | 0.72 | 3.03 | **0.56** | **0.57** | 0.63 | 0.98 | 0.8 | 3.07 | 0.78 | 3.68 | -0.22 | 1.28 | 0.26 | 1.02 | **0.48** | **0.00** | 0.16 | 0.00 | 0.58 | 0.60 |
| AntCircle | 0.58 | 4.90 | **0.40** | **0.96** | 0.54 | 1.78 | 0.58 | 2.87 | 0.65 | 5.48 | 0.00 | 0.00 | 0.17 | 5.04 | 0.20 | 0.00 | 0.13 | 0.02 | 0.45 | 0.40 |
| **Average** | 0.64 | 3.36 | **0.52** | **0.82** | 0.68 | 1.04 | 0.74 | 3.11 | 0.48 | 3.8 | 0.33 | 1.12 | 0.55 | 2.55 | **0.39** | **0.03** | 0.04 | 0.04 | 0.54 | 0.20 |

## A.5 COMPOSITION AND SAFETY PROFILES OF THE DATASETS

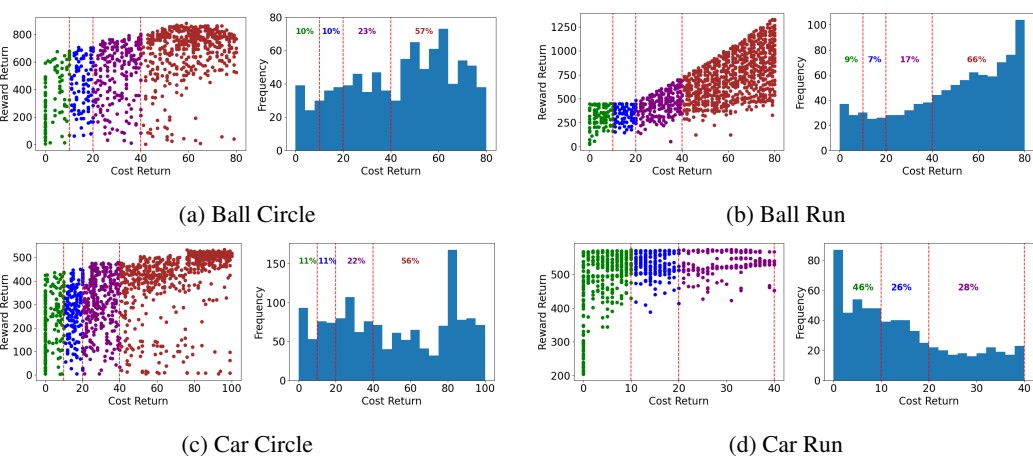

(a) Ball Circle

(b) Ball Run

(c) Car Circle

(d) Car Run

Figure 10: Distribution of the trajectories in the dataset based on cost return and reward return

Almost all of the datasets used in our evaluations include both safe and unsafe trajectories. The safety compliance of policies during evaluation is assessed based on their adherence to the undiscounted cost-return threshold established in our experiments, consistent with standard benchmarks. The red dashed lines in figure 10 indicate these thresholds.

The figures in 10a - 10d include scatter plots showing the reward return versus cost return for trajectories across four representative datasets, including those used in the ablation study. Each point corresponds to a single trajectory/episode. Additionally, the frequency distributions of cost returns for these datasets are provided in the right, highlighting that many trajectories exceed the safety threshold, often with unsafe trajectories outnumbering safe ones. These visualizations illustrate the datasets' composition and demonstrate the robustness of LSPC in learning policies that maintain safety compliance despite mixed safety profiles.

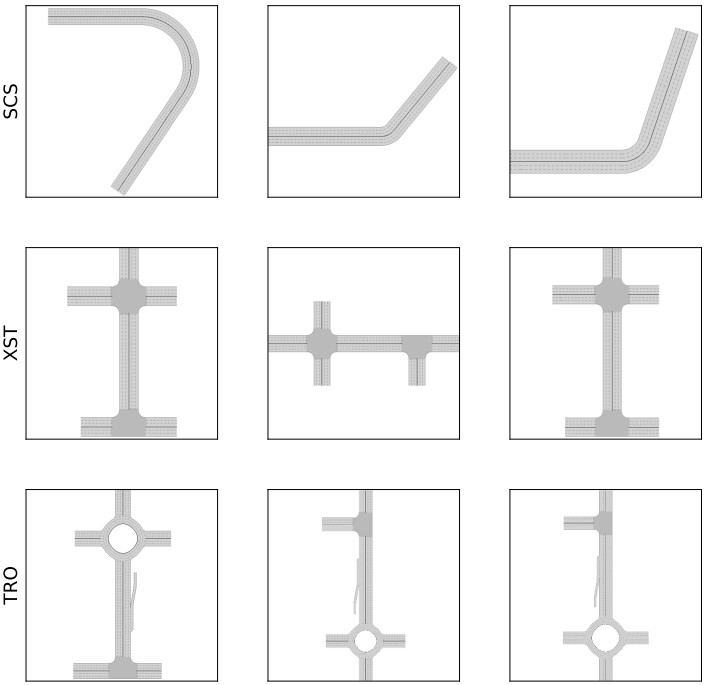

Figure 11: Block sequence and corresponding map samples in the Metadrive Simulator

Table 6: Metadrive Environment Configurations

| Environment | Traffic Density | Block Length | Block Sequence | Horizon | Lane Width | Lane Count |
|---|---|---|---|---|---|---|
| EasySparse | 0.1 | 3 | SCS | 1000 | 3.5 | 3 |
| EasyMean | 0.15 | 3 | SCS | 1000 | 3.5 | 3 |
| EasyDense | 0.2 | 3 | SCS | 1000 | 3.5 | 3 |
| MediumSparse | 0.1 | 3 | XST | 1000 | 3.5 | 3 |
| MediumMean | 0.15 | 3 | XST | 1000 | 3.5 | 3 |
| MediumDense | 0.2 | 3 | XST | 1000 | 3.5 | 3 |
| HardSparse | 0.1 | 3 | TRO | 1000 | 3.5 | 3 |
| HardMean | 0.15 | 3 | TRO | 1000 | 3.5 | 3 |
| HardDense | 0.2 | 3 | TRO | 1000 | 3.5 | 3 |

## A.6 METADRIVE TRANSFER EXPERIMENT

For rigorous evaluation of the proposed methods in varying scenarios, a set of nine distinct environments with different configurations is employed in this study. These environments are categorized into three difficulty levels: Easy, Medium, and Hard. Each difficulty level includes three variations based on traffic density: Sparse, Mean, and Dense. Each environment is characterized by specific parameters, including the start seed, traffic density, block length, block sequence, simulation horizon, lane width, and lane count. The block sequences selected correspond to the difficulty of the driving task: SCS (Straight-Curve-Straight) for easy levels, XST (Intersection-Straight-$T$Intersection) for medium levels, and TRO ($T$Intersection-OutRamp-Roundabout) for hard levels. The configuration is summarized in table 6 and the example maps corresponding to each block sequence type are illustrated in figure 11.

In the transfer experiment, we aim to test the generalization capability of offline RL algorithms by training agents in one environment configuration and then transferring them to a different configuration without retraining. Specifically, we focus on agents trained in two extreme scenarios: Easy Sparse and Hard Dense. We then evaluate their zero-shot transfer performance across a range of other configurations. This setup enables us to assess how effectively the agents can adapt to new

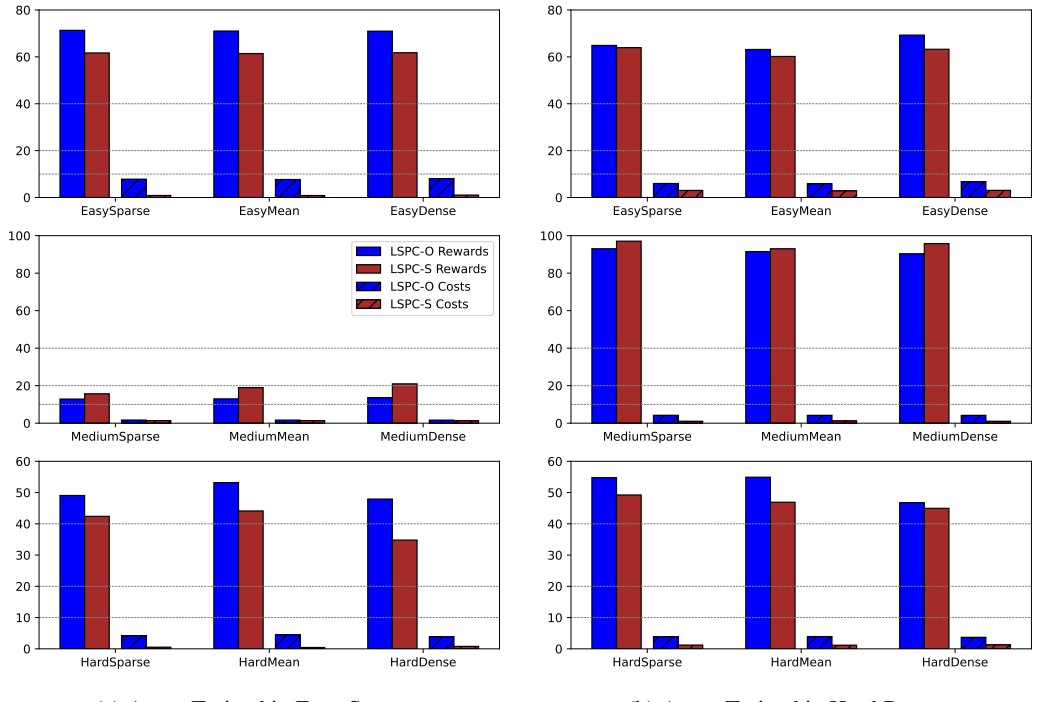

(a) Agent Trained in Easy Sparse       (b) Agent Trained in Hard Dense

Figure 12: Zero-shot transfer performance of agents trained in the Easy Sparse and Hard Dense scenarios when evaluated in various other settings. The unhatched bars represent the average normalized reward return, while the hatched bars indicate the average cost return, evaluated across 20 episodes. The dashed lines represent the different cost threshold used in this paper and in prior works.

environments without additional training, providing insights into the robustness and flexibility of the learned policies.

The results of this experiment are illustrated in figure 12. In the figure, unhatched bars represent the average normalized reward return, while hatched bars indicate the average cost return, both evaluated over 20 episodes. The dashed lines correspond to the different cost thresholds used in this study as well as the DSRL (Liu et al., 2023a). Notably, the agent trained in the Hard Dense scenario exhibited improved reward performance when transferred to easier and sparser tasks, even surpassing its own performance in the Hard Dense environment. This outcome is likely due to the agent's exposure to more challenging data during training, which enhanced its adaptability to simpler settings—a result that is not always commonly observed in transfer learning scenarios. In contrast, the agent trained in the Easy Sparse environment experienced a performance drop when transferred to denser and more complex tasks, with the decline being more pronounced in medium difficulty scenarios than in the hardest ones. Despite these variations in reward performance, both agents consistently maintained safety across all cost threshold levels, demonstrating robust safety across all difficulty and density configurations. This consistency aligns with the core principle of our method, Latent Safety-Prioritized Constraints (LSPC), which ensures that the policy maximizes rewards while adhering to safety constraints within the latent space.

