# OpenReview forum: "Latent Safety-Constrained Policy Approach for Safe Offline Reinforcement Learning"
_ICLR.cc/2025/Conference — ICLR 2025 Poster_

### Official Review · Reviewer_MBPj · 2024-10-31

**Soundness:** 3
**Presentation:** 2
**Contribution:** 3
**Rating:** 6
**Confidence:** 4

**Summary:**

This paper develops a novel approach to safe offline reinforcement learning; a task that aims to maximize rewards and minimise safety constraint violations during training on an offline dataset. The proposed methodology leverages conditional variational autoencoders (CVAEs) to model latent safety constraints, effectively learning safety bounds. The proposed methodology can then be encompassed into two further methodologies that the authors evaluate upon: LSCP-S which prioritizes selecting the safest actions, and LSPC-O, which aims to maximize the reward within these latent safety bounds. In the evaluation, it is seen that LSCP-S effectively reduces cost and LSCP-O manages a nice balance between reducing cost and maximising reward.

**Strengths:**

- A promising approach to capture complex, non-linear safety constraints. Results clearly back this up and I was impressed with the methodology and theoretical analysis.
- The literature review is comprehensive and up-to-date.
- The broad range of benchmarks that the methodology is evaluated upon is robust and impressive.
- Authors are transparent about the methodology's limitations and future work, showing openness and foresight.
- Figure 3, in particular, is visually engaging and provocative for the reader. It enables further transparency into the proposed method and I really appreciated it.
- Code is available and open.

**Weaknesses:**

- There is little to no discussion, apart from minor comments in the related work section, regarding comparative methods and hyperparameters which strongly hinders the assessment of Table 1. For example, in the metadrive experiments, CPQ doesn't seem to learn anything. More detail into this method or the hyperparams used at the start of the the evaluation section could have provided me with further insight. Currently, I am left thinking that the implementation wasn't tuned correctly. Also, there is no standard RL baseline against which to assess all methods.
- An analysis point, I would like/liked to have seen a discussion on surrounds the comparison between LSPC-S and LSPC-O. In Table 1, there seems to be a disproportionate increase in cost between the -S and -O models. I would have expected, within reasonable deviation, to have seen a reasonably proportionate increase in reward to increase in cost between the -S and -O models. But, in the hard metadrive experiments, for example, the reward increased +0.03 on average whilst the cost increased +0.24 on average. I am not concerned at all about this, I would just like to see some discussion surrounding this interesting trade-off that the -O model has to try to balance.
- Another analysis point, which slightly draws parallels from my previous point, is in Figure 2 it seems that LSCP-O observes large oscillations in terms of cost in comparison to other methods. The small increase in reward is seen from the -S to the -O model but a larger cost is also observed alongside some stronger oscillation. Could this oscillation be due to the model trying to find a trade-off balance or is this some hyperparam tuning that is needed?
- Computational/Time Taken/Machine experimented on details are needed in the appendix. Without these, I personally would draw the conclusion that the proposed methods are computationally more intensive than comparative methods. Personally, I do not computation as a major weakness, but I would appreciate details about it to be transparent.

Minor weaknesses
======================
- Figures 2 and 3 have very small text and need to be increased. The legends are also in a poor place, personally, and could be better placed above/outside the axis.
- Tables could also benefit from rows having alternating highlights (white to light gray) to make them easier to read.
- Please put boldface and blue colour reasoning in the table caption for readability.

**Questions:**

Can the authors comment on the following:
- The disproportionate increase in rewards to the large increase in costs from the -S to -O models. A point that is then observed again in Figure 2.
- Are the oscillations observed in Figure 2 a minor weakness with the method or simple hyperparam tuning that is needed?
- Can you add experiment insights (computation/time taken/machine experimented on) into the appendix to improve transparency?

---

> ### Author Response · Authors · 2024-11-20
> **Response to the reviewer MBPj's comments**
>
> Thank you for your insightful review and encouraging feedback on our work. We are pleased that our empirical and theoretical contributions were well-received and found promising. We really appreciate your recognition of the paper's strengths. In the following paragraphs, we address your comments and questions in detail.
>
> **Baselines and their hyperparameter tuning**
>
> To clarify, the CPQ results in Table 1 are based on the original results reported in the DSRL whitepaper, some of which we verified by running CPQ ourselves, yielding comparable results (as shown in Figure 2b). Recent works, such as FISOR, also report similar CPQ performance, further supporting the results provided here.
> Regarding the reviewer’s concern about standard RL baselines, FISOR is a robust recent baseline that serves this purpose effectively in our comparisons. We agree that detailing the hyperparameters for all baselines in the main text would enhance clarity; however, given space constraints, we propose adding these details to the supplementary materials or appendix and referencing the corresponding works/repos for further implementation specifics.
>
> **Safety-performance trade-off in LSPC-S vs LSPC-O**
>
> We appreciate the reviewer's insightful comment regarding the comparison between LSPC-S and LSPC-O, particularly the disproportionate increase in cost relative to reward in Table 1. In the hard metadrive experiments, we conducted hyperparameter tuning for LSPC-O to assess potential performance improvements, including increasing the size of the restricted latent safety space and raising the dimensionality of the latent space from 32, as we suspected that this might not be expressive enough for the task's difficulty. However, we found that these adjustments resulted in only increased costs, while reward performance tended to saturate, likely due to the challenging nature of the tasks. Consequently, efforts to enhance rewards often led to similar disproportionately larger increases in cost, as noted by the reviewer. Although our methods significantly outperform other baselines in this set of tasks, we acknowledge that the reviewer’s observation may also reflect some limitations inherent in our proposed framework.
>
>
> **Oscillation of cost return while training LSPC-O**
>
> This is an astute observation. While there may be alternative sets of optimal hyperparameters, we opted to use a consistent set across all tasks. We believe that the oscillations are primarily attributed to the model's attempts to find a balance between safety and reward optimization, as you noted.
>
> Firstly, during early training stages, the CVAE decoder, being a neural network, is still undertrained and has limited exposure to samples from the prior distribution. While capable of modeling complex, multimodal distributions of in-distribution actions, the decoder may initially map some samples to out-of-distribution or unsafe actions. This occurs as the latent safety encoder ($\delta$) seeks to maximize rewards by exploiting the inaccuracies in these mappings. Over time, as the decoder encounters a more comprehensive set of samples from the prior, it stabilizes, forcing the encoder to select actions from safer, in-distribution regions.
>
> Secondly, LSPC-S is designed to be conservatively safe. When the latent safety encoder in LSPC-O regresses over this latent space, our findings show that it tends to select actions that are on the higher end of the cost Q-value spectrum. This means that among the actions modeled by the LSPC-S decoder, LSPC-O often chooses actions that correspond to higher cost Q-values. This behavior is intuitive, as LSPC-S represents a conservative policy derived from the offline dataset, and any other policy (including LSPC-O) seeking reward maximization may not achieve the same level of safety.
>
> We think that these factors mainly contribute to the oscillating and higher costs observed for LSPC-O during training.
>
> ***Page (1/2)***

---

> > ### Author Response · Authors · 2024-11-20
> > **Response to the reviewer MBPj's comments (continued)**
> >
> > **Computational time and experiment details**
> >
> > The reported training times are for experiments conducted on a Dell Alienware Aurora R12 system with an 11th gen Intel Core i7 processor, 32 GB DDR4, and an NVIDIA GeForce RTX 3070 8GB GPU.
> >
> > Running on a CUDA device, LSPC training takes approximately 3.5 hours for 1 million training steps, achieving a rate of around 90 training iterations per second. It is important to note that this reported training time incorporates the training of both LSPC-S and LSPC-O components of our framework. This performance is comparable to CPQ under similar conditions and training steps. BC, being more straightforward, is faster, requiring about 35 minutes for 1 million timesteps at 490 training steps per second. CDT training for 100k timesteps takes approximately 3.5 hours, running at about 8 iterations per second.
> >
> >
> > **Minor Weaknesses**
> >
> > Thank you for your thoughtful suggestions; we appreciate the reviewer’s detailed review of the paper. We will incorporate the recommended changes in the revised manuscript to enhance the readability of the figures and the clarity of the tables.
> >
> >
> >
> > ***Page (2/2)***

---

> > > ### Comment · Reviewer_MBPj · 2024-11-22
> > >
> > > > Baselines and their hyperparameter tuning
> > >
> > > After reviewing the whitepaper, I now see that this is indeed the case. Thank you for highlighting it. For clarity, you might consider explicitly stating (perhaps around lines 469/470) that this aligns with the expected results. Additionally, including the reference again would ensure your findings remain clear and accessible to new readers who might initially have the same thoughts as I did.
> > >
> > > Adding details surrounding hyperparameters in the appendix is more than sufficient, thank you.
> > >
> > > > Safety-performance trade-off in LSPC-S vs LSPC-O
> > >
> > > I will clarify, that I had no concern about the proposed method's validity or performance regarding this trade-off. I believe it would raise an interesting discussion that could potentially be included in Appendix A.4.
> > >
> > > > Oscillation of cost return while training LSPC-O
> > >
> > > Thank you for the discussion surrounding this point.
> > >
> > > > Computational time and experiment details
> > >
> > > Thank you, including these in the Appendix would be beneficial for reproducibility.

---

### Official Review · Reviewer_Qh7M · 2024-11-01

**Soundness:** 3
**Presentation:** 3
**Contribution:** 2
**Rating:** 6
**Confidence:** 4

**Summary:**

Traditional RL learns optimal policies through interactions with the environment, but this approach can pose safety risks during the learning process. In industrial settings such as autonomous driving and robotics, even minor errors can lead to significant costs or safety hazards. To address these challenges, offline reinforcement learning has been introduced. However, offline RL, which relies solely on fixed datasets, risks generating unsafe policies when encountering new situations. This study proposes a new framework called Latent Safety-Prioritized Constraints (LSPC), designed to learn offline policies that optimize rewards while adhering to safety constraints, demonstrating its effectiveness across various high-risk tasks.

**Strengths:**

1.  Balance of safety and performance: Designed to optimize rewards while satisfying safety constraints.
2.  Efficiency of offline learning: Learns policies without further interactions with the environment by using fixed datasets.
3.  Theoretical stability guarantee: Offers theoretical bounds on policy performance and sample complexity.
4.  Applicability in various environments: Demonstrates strong performance in complex tasks, including autonomous driving and robotic manipulation.

**Weaknesses:**

1.  Difficulty in latent space configuration: Finding the optimal latent space configuration requires empirical tuning.
2.  Dependency on dataset quality: Safe policy learning heavily relies on the quality of the dataset.
3.  Limitations in real-time application: Offline RL struggles to respond immediately to dynamic changes in real-time environments.

**Questions:**

## Theoretical Concerns

1.  Setting safety constraints in the latent space

    LSPC uses a conditional variational autoencoder (CVAE) to learn safety constraints in the latent space. However, it is unclear whether the latent space can fully capture real-world safety constraints. If the safety constraints are not properly learned, the model may fail to ensure safety in unforeseen situations.

2.  Reliability of constrained optimization

    LSPC performs constrained reward optimization to maintain a balance between rewards and safety. However, it raises concerns about whether the optimization in the latent space will remain valid for actions not encountered during training. The paper lacks guarantees that the policy will behave appropriately for out-of-distribution state-action pairs.

3.  KL divergence-based safety assurance

    While safety is ensured by minimizing the Kullback-Leibler (KL) divergence between the learned policy and behavior policy, subtle behavioral differences, even with small divergences, could lead to significant risks in complex environments. Theoretical minimization of KL divergence may not fully capture all possible real-world scenarios.


----------

## Experimental Concerns

1.  Gap between simulation and real-world environments

    The paper evaluates performance primarily in simulated environments. However, simulations may not fully account for real-world uncertainties, such as noise, sensor errors, or dynamic changes. This gap poses a risk that LSPC’s safety and performance may fall short in real-world applications.

2.  Dataset limitations and bias

    Since LSPC relies on static datasets, there is a risk that the policy will not perform well in situations not included in the dataset. Furthermore, if the training data is biased toward specific patterns, the policy may make incorrect decisions.

3.  Interpretation of safety-performance trade-offs

    While the paper claims to maintain a balance between safety and rewards, it may not have thoroughly analyzed cases where this balance breaks down. For example, certain tasks may allow minor safety violations to achieve higher rewards, but the paper lacks a clear discussion of these scenarios.

4.  Stability of optimization

    The complex optimization process involving CVAE and implicit Q-learning can lead to convergence issues, resulting in unstable learning or local optima. The paper does not provide adequate solutions to address these potential issues.


----------

## Comprehensive Concerns

-   Are the safety constraints in the latent space sufficient? It remains unclear whether CVAE can effectively capture all necessary safety constraints.
-   Lack of generalization to real-world scenarios: Reliance on static datasets raises concerns about performance degradation in new situations.
-   Trade-offs between safety and performance: While the framework claims to balance safety and rewards, it may fail to guarantee absolute safety in specific scenarios.
-   Complexity of the optimization process: Additional analysis is required to address potential convergence issues and instability during learning.

These concerns highlight important considerations for evaluating the applicability and reliability of LSPC. Future research should focus on validating its generalization performance in real-world environments and ensuring that the balance between safety and rewards is consistently maintained across various scenarios.

---

> ### Author Response · Authors · 2024-11-20
> **Response to the reviewer Qh7M's comments**
>
> Thank you for your thoughtful review and feedback. We are glad to know that you appreciated the balance between safety and performance, as well as the theoretical analysis in our work. In the following paragraphs, we will try to answer your concerns in detail.
>
>
> **Theoretical Concerns**
>
>
> **Capturing real world safety constraints in the latent space**
>
> The CVAE in LSPC is designed to capture safety constraints by learning a distribution of safe actions conditioned on observed states, thereby effectively modeling safety within the dataset's scope. While any data-driven model may face limitations in unforeseen situations, future work could address this by incorporating uncertainty estimation or ensemble methods to identify out-of-distribution actions. Within our experiments, the approach demonstrates substantial empirical effectiveness in benchmark simulation environments. Although real-world tests were not conducted in our study, we evaluated robustness by transferring policies across MetaDrive tasks (Appendix A.5) and found that the policies maintained safety constraints, suggesting reliable constraint adherence even under varying task scenarios.
>
>
> **Reliability of constrained optimization and generalization beyond the dataset**
>
> In offline RL, selecting in-distribution actions is critical to maintaining reliability, as the agent is limited to the behaviors observed in the dataset and cannot explore. LSPC addresses this by optimizing within a CVAE-trained latent space that models safe, in-distribution actions, helping to ensure that policy actions adhere to the dataset’s safety constraints. This design reduces the risk of unsafe or unpredictable behaviors for out-of-distribution states. Notably, this selection of in-distribution actions is especially crucial in safe offline RL, where unseen actions may pose significant risks of constraint violations that cannot be assessed due to the lack of interaction.
>
> Additionally, the policy is learned through training a parameterized deep learning model, which has been shown to have good generalizability despite its possible algorithmic instability, nonrobustness, and sharp minima (Kawaguchi et al. 2017). Therefore, even for state-action not encountered during training, the optimization in the latent space will remain valid if data distribution drift is not extreme, as CVAE should, to some extent, be able to handle unseen actions during training. In an extreme scenario where most state-action pairs are out-of-distribution, an online fine-tuning method can be adopted quickly to ensure our method to be performant. Alternatively, a robust variant of our proposed algorithm can also be developed to mitigate this issue.
>
>
> **KL Divergence Minimization Between Learned and Behavior Policies and Its Implications**
>
> We thank the reviewer for raising such a good point for our theoretical analysis. Indeed, the KL divergence may not always accurately represent the complexities of real-world situations, but it has widely been used in many offline (safe) RL methods, as it is able to prevent drastic deviations from the behavior policy. To enhance the applicability in real-world scenarios, it is crucial to carefully design a behavior policy that captures the desired behaviors and safety constraints a priori. Regardless of the gap between theory and practice, the minimization of KL divergence is still an effective approach to approximately capture the possible real-world scenarios. Also, KL divergence as a regularizer prevents the learned policy from making sudden large jumps away from the behavior policy, leading to more stable learning and smoother policy updates. It also encourages the exploration-exploitation balance by limiting how much the learned policy can diverge from the behavior policy and prioritizing actions that are likely to be successful based on the observed data. It is also a decent metric to handle complex environments where the optimal policy might not be easily identifiable, since the agent learns a policy that is still reasonably close to the observed behaviors.
>
> ***Page (1/3)***

---

> ### Author Response · Authors · 2024-11-20
> **Response to the reviewer Qh7M's comments (continued)**
>
> **Experimental Concerns**
>
> **Gap between simulation and real-world application**
>
> We acknowledge that simulations may not fully capture real-world uncertainties and could impact the performance of LSPC and any other reinforcement learning method in practical applications. While our main contribution is a general framework that should be applicable to real datasets as well, we have opted to use established simulated environments for comparison with existing baselines to ensure consistency. We recognize that bridging the sim-to-real gap is indeed a challenge and an active area of research as well.
>
> To extend the learned policy from a simulation environment to a real-world deployment, we can calibrate the simulation environment with prior domain knowledge from a real test-bed or environment such that the learned policy can be more adaptive. Other techniques from sim-to-real literature can also be utilized to ensure the latent space to fully capture real-world safety constraints, which can indeed be an interesting direction. Nevertheless, as noted in the conclusion section as a future work, we are committed to exploring the application of LSPC in real-world settings to validate its effectiveness under various uncertainties and dynamic conditions and testing its potential capabilities in sim-to-real applications.
>
>
>
>
> **Dataset limitations and bias**
>
> Indeed, the reliance on pre-collected data is a fundamental characteristic of offline RL methods, including ours. We acknowledge that this approach may lead to suboptimal performance in situations not represented in the dataset or when the training data exhibits bias towards specific patterns. However, it is important to note that this challenge is not unique to LSPC but is inherent to all offline RL methods. In fact, regularizing the learned policy to remain close to the behavior policy is a common and necessary practice in offline RL to address the issue of distribution shift. While this constraint may limit the policy's ability to generalize to entirely novel situations, it is a crucial safeguard against the potential unsafe behavior that can arise from extrapolating beyond the known data distribution. We appreciate this feedback and recognize the importance of addressing OOD generalization in offline RL.
>
>
> **Interpretation of safety-performance trade-offs**
>
> We agree that certain tasks may allow minor safety violations to achieve higher rewards. In fact, our benchmark tasks, as detailed in Appendix A.4, allow different levels of constraint violation cost budget for each trajectory. While our method does not directly operate on this budget, we use a restriction hyperparameter $\epsilon$ that controls how restrictive the latent space is for the optimal policy search. This hyperparameter effectively manages the safety-performance trade-off.
>
> As illustrated in Figure 4, varying levels of $\epsilon$ can change both safety adherence and reward performance. Increasing $\epsilon$ allows the latent safety encoder policy in LSPC-O to explore a larger region within the latent space, potentially finding actions that are better at reward optimization. However, this improved performance comes with a trade-off of higher cost returns. However, this improved performance comes with the trade-off of higher cost returns. As the reviewer noted, policies can be considered 'optimal' for different levels of allowed cost budgets, but when the search space becomes excessively large, this balance may break down, leading to unsafe and unrecoverable behaviors.
>
> As an additional example/ablation, here [ https://imgur.com/a/i4jXnsc ] are the results of training our methods with different levels of restriction on the halfcheetah-velocity environment. In this specific example, the cost return of LSPC-O increases beyond the standard thresholds considered in this work when $\epsilon$ is 0.5 or more. However, LSPC-S which we can always obtain as a backup policy during the LSPC-O training maintains the safety adherence, as observed in the second figure in [ https://imgur.com/a/i4jXnsc ].
>
> To summarize our answer, we provide a flexible framework that can be tuned to different safety requirements while still optimizing for rewards within those constraints.
>
> ***Page (2/3)***

---

> > ### Author Response · Authors · 2024-11-20
> > **Response to the reviewer Qh7M's comments (continued)**
> >
> > **Stability of optimization**
> >
> > We really appreciate these insightful comments on optimization and learning stability. For the convergence, as we still leverage the first-order stochastic optimization algorithms such as Adam or AdamW in the implementation, the convergence rate is sublinear, typically in an order of $\mathcal{O}(\frac{1}{\sqrt{T}})$, where T is the number of iterations, as the problem is an regular non-convex optimization problem. This has been well-studied in the stochastic optimization area. Even if the optimization is dynamic, we can also use the metric of regret to show the similar conclusion. We will add a technical discussion and cite some references in the revised draft. Since it is a non-convex optimization, the convergence is only to a first-order stationary point, which has popularly been recognized as a locally optimal solution, unless with some assumptions for the objective function, such as Polyak-Lojasiewicz (PL) condition, it can be shown that the convergence is faster than $\mathcal{O}(\frac{1}{\sqrt{T}})$. For the unstable learning, in our work, we have used the experience replay buffer and target networks in IQL, which are widely used techniques in the RL community to address the issue. Additionally, KL divergence as a regularizer prevents the learned policy from making sudden large jumps away from the behavior policy, leading to more stable learning and smoother policy updates. The soft updates and learning rate decay are applied as well to mitigate the learning instability.
> >
> >
> > **Comprehensive Concerns**
> >
> > We appreciate your comprehensive concerns and believe we have already addressed most of them in previous sections. Below are brief responses to further clarify our position:
> >
> > *1. Sufficiency of Safety Constraints in the Latent Space:* While it is (a) challenging to theoretically guarantee that the latent space fully captures all necessary safety constraints, and (b) a fundamental limitation of any data-driven method relying on an offline dataset, our empirical results demonstrate strong safety adherence when optimizing within the latent safety boundaries inferred by the CVAE.
> >
> > *2. Generalization to Real-World Scenarios:* We agree that relying on static datasets can raise concerns regarding performance degradation in new situations. While our study focuses on simulated environments, we have discussed potential solutions such as calibration with real-world domain knowledge and techniques from the sim-to-real literature.
> >
> > *3. Trade-offs between safety and performance:* We recognize that the safety-performance trade-off is a critical concern. Our framework offers a flexible approach, with hyperparameters such as $\epsilon$ to adjust the balance between safety and reward maximization. We have shown, with some examples, that this balance can be adjusted according to specific task requirements.
> >
> > *4. Complexity of the Optimization Process:* Regarding convergence and optimization stability, we have discussed the use of first-order stochastic optimization methods, such as Adam, along with regularization techniques like KL divergence, which help prevent instability and drift in policy learning.
> >
> > We acknowledge the importance of these comments for assessing the applicability and reliability of LSPC. And as you rightly pointed out, exploring its generalization to real-world environments can be a key direction for future research.
> >
> > ***Page (3/3)***

---

> > > ### Comment · Reviewer_Qh7M · 2024-11-25
> > > **Official Comment by Reviewer Qh7M**
> > >
> > > Thank you for taking the time to provide thoughtful feedback. The theoretical concerns raised have been addressed through the responses and were sufficient to provide clarity. While the experimental and comprehensive concerns were not directly addressed, they were adequately resolved within the context of the theoretical explanations. As such, the original score will remain unchanged.

---

### Official Review · Reviewer_NRFb · 2024-11-03

**Soundness:** 3
**Presentation:** 3
**Contribution:** 2
**Rating:** 6
**Confidence:** 3

**Summary:**

This paper presents a novel Latent Safety-Prioritized Constraints (LSPC) framework aimed at addressing the balance between reward maximization and safety constraints in safe offline reinforcement learning. The authors model potential safety constraints using Conditional Variational Autoencoders (CVAE) and demonstrate experimental results across multiple benchmark tasks, indicating that the proposed method effectively enhances cumulative rewards while ensuring safety.

**Strengths:**

Originality:

The LSPC framework offers a fresh perspective on managing the trade-off between safety and rewards, particularly within the context of offline reinforcement learning, demonstrating significant practical and theoretical value.

Significance:

The paper provides a detailed and persuasive theoretical analysis of policy performance, ensuring the scientific foundation of the proposed method.

Quality:

Through systematic experimentation across various benchmark tasks, the effectiveness of LSPC is showcased, with significant advantages over existing methods.

**Weaknesses:**

Please refer to questions

**Questions:**

- Although the experiments primarily focus on specific tasks, there is a lack of in-depth discussion on the application and effectiveness of the LSPC framework in multi-task learning. How will you address the variations in safety constraints and reward structures across different tasks?

- When comparing with several benchmark methods, the analysis does not adequately demonstrate the specific advantages of LSPC across different tasks. Is it necessary to provide a more detailed discussion of the comparison methods, especially regarding their performance under different safety constraints?

- When using Conditional Variational Autoencoders (CVAE) to construct a latent safe space, how do you ensure that the generated samples comprehensively cover all possible safety constraints in the environment? Is it necessary to validate the effectiveness and sufficiency of this coverage?

- Decoupling reward maximization and safety constraints may lead to failures in safety assurance. How does the decoupled strategy effectively guarantee safety under different safety constraint conditions?

- The thoretical analysisi is dense, it's better to provide more context and examples to help readers understand the impact and theoretical basis of this assumption.

---

> ### Author Response · Authors · 2024-11-20
> **Response to the reviewer NRFb's comments**
>
> Thank you for the thorough review of our work. We are grateful that you found our approach to balancing safety and reward maximization with a generative model in offline safe reinforcement learning to be an innovative contribution. We also appreciate your recognition of the theoretical foundation and strong empirical results in our work. In the following paragraphs, we will try to answer your review comments and rebuttal questions.
>
> **Effectiveness and application in multitask learning**
>
> We agree that multi-task reinforcement learning represents an exciting direction for extending the LSPC framework. Such an extension could examine whether the inferred latent safety boundary might serve as a shared safety constraint representation across tasks. However, as of right now, applying the framework across tasks with differing observation and action space dimensions could introduce additional challenges and may require some innovation in the current architecture.
> In Appendix A.5, we conduct an exploratory analysis of policy transfer across the MetaDrive tasks, observing that policies retain safety when transferred, which we find promising. Your suggestion to investigate variations in safety constraints and reward structures as a broader multi-task application aligns well with our findings and could form an intriguing future line of research. Nevertheless, we want to emphasize that our main contribution and novelty is in safe offline reinforcement learning; this focus on single-task scenarios allows us to thoroughly address the core challenges of safe offline RL, which remain relevant and crucial in learning safe policies from static datasets.
>
>
> **Handling variations in safety constraints and reward structures across tasks**
>
> Across different tasks, we can extend the single encoder-decoder model to multiple encoder-decoder models that are able to learn safety constraints tailored to each individual task. Alternatively, we can design hierarchical constraints, which define high-level safety principles that are then transferred into specific constraints for each task. Though the current problem formulation is able to take different safety constraints into account, we have primarily focused on a single task. The extension could be fairly non-trivial in terms of algorithm design. Additionally, due to the diverse natures of reward structures, reward engineering also plays a central role in efficient learning. Some weighting techniques may be required for each reward component. Another appealing method is to establish the multi-objective optimization problem for different tasks and to search for the optimal Pareto front, which presents a set of solutions that are non-dominated to each other but are superior to the rest of solutions in the search space. We believe that addressing the variations in safety constraints and rewards structures across different tasks is of indispensable importance and independent interest, and can be a critical future research direction.
>
> **Specific advantages of LSPC across different tasks and performance under different safety constraints**
>
> Our approach with LSPC is designed to learn safety constraints from the dataset and represent them in a latent space, allowing for generalized handling of constraints regardless of their specific nature. While we have provided detailed descriptions of tasks and safety constraints in Appendix A.4, we acknowledge that including more details on different safety constraints in the main text would enhance the paper's self-containment and accessibility. Additionally, Figure 3 attempts to visualize the constraints represented by the inferred latent safety boundary in the action space, offering some insight into how LSPC interprets and enforces these constraints.
>
> ***Page (1/2)***

---

> > ### Author Response · Authors · 2024-11-20
> > **Response to the reviewer NRFb's comments (continued)**
> >
> > **Comprehensive coverage of safety constraints in generated samples**
> >
> > Our approach, being data-driven (i.e. *offline*), is inherently limited by the coverage of the constraint violation signals in the static dataset. The CVAE policy is designed to retrieve and represent these signals in a more tractable and enforceable form, facilitating safe policy training. While comprehensive coverage of all possible safety constraints is largely dependent on the dataset quality, our method captures long term constraint violation by learning $V_c$ using IQL from the constraints present in the training data and enforces it during CVAE training. We acknowledge that out-of-distribution generalization remains an open challenge in offline learning, and incorporating domain knowledge or prior information about constraints could be beneficial in such cases. We (partly) answer this question in terms of sample complexity analysis in appendix A.2. This analysis offers insights into the theoretical bounds of our method's performance given the available data. While this does not guarantee complete coverage, it provides a framework for understanding the relationship between data availability and the effectiveness of learned safety constraints.
> >
> >
> > **Ensuring safety under varying constraints with the decoupled strategy**
> >
> > The constrained optimization problem posed in Equation 2 involves optimizing over all possible policies in the policy space and is indeed challenging to solve directly. Most existing safe offline RL methods, including ours, approximate solutions to this broad problem. While it is true that we decouple cost minimization and reward maximization, we do so with the intent of deriving a conservative policy represented by the CVAE decoder. Specifically, for latent variables that have a high probability under the prior distribution (assumed to be unit Gaussian), the decoder $p_{\beta}(a \mid s, z)$ is expected to produce actions with high likelihood under the conservatively safe policy distribution $\pi_s(a \mid s)$. In LSPC-S, we sample from this high-probability region, whereas LSPC-O involves an encoder trained to seek reward maximization while remaining in the same constrained region. By focusing on actions generated from this safety-prioritized latent space, we ensure that both methods adhere to the safety constraints in practice.
> >
> >
> > **More context and examples on theory and related assumptions**
> >
> > We provide an overview of theory and related assumptions in this context to help readers understand the impact. In summary, the theoretical analysis in our work can be split into two parts: the first one includes the performance gap driven by the reward maximization, and the constraint violation bound due to the safety constraints; the second part consists of the sample complexity analysis, which has stayed in the Appendix. However, we will bring key statements back to the main contents, suggested by a reviewer. Since it is one of the main theoretical contributions. Specifically, the first part intuitively tells us how large the gap is between the optimal and learned policy. This gap is dictated by the maximum of immediate reward and the upper bounds of difference among the safe policy, learned policy and optimal policy, in Assumption 1 and Assumption 2. These two assumptions enforce constraints that the difference among the three policies should be bounded above, instead of being infinity. Such assumptions also implicitly ensure learning stability. In practice, we can use some hyperparameters such as learning rate to control the learning progress to provide the assurance of stability. The constraint violation bound indicates that when the constraint is violated (as we have implemented a soft constraint, instead of a hard constraint), the violation has its largest value, instead of infinity, which would directly fail our proposed framework.
> > However, since the policy is learned from an offline dataset, would increasing the size of the dataset reduce the gap and the bound in our first part of analysis? The second part of analysis attempts to answer this question. We would like to see how the performance gap and the constraint violation bound decay along with the size of the offline dataset. We impose two assumptions, one on the KL divergence between the learned policy and the optimal policy and another on the quality of the offline dataset. Since, in our study, we use CVAE to learn the latent safety constraint from the data, which requires the dataset to at least have some data generated by safe policies. With these two assumptions in hand, we provide two main results in the sample complexity, yielding the decay rates of the performance gap and constraint violation bound along with the size of the offline dataset.
> >
> > ***Page (2/2)***

---

> > > ### Comment · Reviewer_NRFb · 2024-11-26
> > >
> > > Thank you for your response. I believe most of my questions have been well addressed. However, I still find it challenging to connect the theoretical analysis with the proposed method. While I do not object to the acceptance of this work, I will maintain my current score.

---

> > > > ### Author Response · Authors · 2024-11-26
> > > >
> > > > We appreciate your feedback and are grateful for your affirmation of the work’s acceptance. We are pleased to hear that most of your questions have been addressed and welcome the opportunity to further clarify the connection between the theoretical analysis and the proposed method.
> > > >
> > > > In LSPC, we leverage CVAE to reconstruct safe policies residing in the offline dataset $\mathcal{D}$, strategically ensuring that the further optimal solutions are safe. This justifies Assumption 1 and Assumption 2 that provide new upper bounds for the distribution distances between $\pi$ and $\pi_s$, and $\pi_s$ and $\pi^*$. Previously, instead of $\pi_s$, it was the unknown behavior policy $\pi_b$ [Yao, et al. 2024], which has less assurance of safety than $\pi_s$ and results in relatively looser bounds. Moreover, in LSPC, we adopt the IQL and AWR for critic model updates and policy extraction, which motivates us to use the value function as the metric to evaluate the proposed algorithm. As the policy is learned via a static offline dataset, the performance gap and constraint violation are assessed based on the stationary distributions, instead of a dynamic metric such as regret [Zhong and Zhang 2024]. Theorem 1 and Theorem 2 suggest the performance gap and constraint violation during learning are upper bounded by constants that are correlated with the new distribution distance bounds induced by the safety policy $\pi_s$. Additionally, as the policy evaluation and extraction steps are all parametric with deep learning models, the number of data samples plays a central role in controlling the performance, which has been a well-established fact in modern deep learning community.
> > > > In offline RL, this motivates the need to assess performance gap and constraint violations with respect to the size of $\mathcal{D}$. Thereby, Theorem 3 and Theorem 4 imply the decay rate with respect to the size of $\mathcal{D}$. Though the analysis can apply to related algorithms with some adaptation, the constants in these results, particularly $\varepsilon_1'$ and $\varepsilon_2'$, are uniquely defined, and thus the theoretical conclusions are tailored to the proposed algorithm.
> > > >
> > > > We are revising the manuscript to incorporate all major comments from the reviewers, including this clarification, and will upload it soon. If this explanation addresses your concern, we hope you might reconsider your evaluation of our work. We would also greatly appreciate any specific suggestions you may have on how to further improve the connection or enhance the work in general.
> > > >
> > > >
> > > > **References**
> > > >
> > > > Yao, Y., Cen, Z., Ding, W., Lin, H., Liu, S., Zhang, T., ... & Zhao, D. (2024). Oasis: Conditional distribution shaping for offline safe reinforcement learning. arXiv preprint arXiv:2407.14653.
> > > >
> > > > Zhong, H., & Zhang, T. (2024). A theoretical analysis of optimistic proximal policy optimization in linear markov decision processes. Advances in Neural Information Processing Systems, 36.

---

### Official Review · Reviewer_He6z · 2024-11-04

**Soundness:** 2
**Presentation:** 2
**Contribution:** 3
**Rating:** 6
**Confidence:** 3

**Summary:**

This paper aims to solve the offline safe RL. To balance safety constraints and distribution shifts in offline safe RL, they use the CVAE to model the policy. Since the CVAE is trained on the offline dataset, this paper expects the CVAE architecture to achieve a balance between safety constraints and distribution shifts. To ensure the CVAE achieves such balance, they propose two methods to use the CVAE. The first method, LSPC-S, penalizes the high-cost state based on the cost function learned by IQL. The second method,  LSPC-O, focuses on reward-maximizing under the CAVE-based policy. This paper demonstrates the effectiveness of their methods through empirical evaluation and theoretical analysis.

**Strengths:**

- The authors provide a theoretical analysis of their method.
- The proposed method outperforms baselines in safe RL tasks.

**Weaknesses:**

- The method proposed by this paper cannot strictly keep safe constraints. Since a state-action pair is safe iff $ V_c^\pi(s)\leq \kappa $. Equation 10 only serves as a regularizer to penalize the high-cost actions and cannot guarantee $ V_c^\pi(s)\leq \kappa $.
- Theory insights are one of the contributions of this work. However, the authors do not derive an exact sample complexity; they only claim that the sample complexity for obtaining an $ \epsilon $-optimal policy is about $ \mathcal{O}(1/\epsilon^4) $and put the sample complexity results in the appendix.
- Lack of ablation study.

**Questions:**

1. Why choose CVAE? Many other generative models, such as the flow and diffusion models, have been shown to outperform VAEs in many fields. Could other generative models achieve better results?

2. Regarding Lemma 2. Theorem 2 in Cen 2024 [1] is bounded by $ \frac{2R_{max}}{(1-\gamma)}D_{TV}[d^D(s)||d^*(s)]+e_N $. I am unable to understand why Lemma 2 is tighter than Cen 2024 since the relationship between $ D_{TV}[d^D(s)||d^*(s)]+e_N $and $ E_{s\sim d^{\pi^*}}[D_{TV}(\pi|\pi^*)] $is unclear. And it's surprising that the upper bound for safe offline RL is tighter than that for offline RL, as offline Safe RL is a more challenging task.

3. I noticed that this paper is similar to [2], as both use CVAE to model policies. What is the difference between the two?

References:

[1] Cen, Zhepeng, et al. Learning from Sparse Offline Datasets via Conservative Density Estimation.

[2] Xu, Haoran, Xianyuan Zhan, and Xiangyu Zhu. "Constraints penalized q-learning for safe offline reinforcement learning." Proceedings of the AAAI Conference on Artificial Intelligence. Vol. 36. No. 8. 2022.

---

> ### Author Response · Authors · 2024-11-20
> **Response to the reviewer He6z's comments**
>
> Thank you for your thoughtful review and helpful suggestions. We are glad you found our approach effective across benchmarks and our theoretical analysis a meaningful contribution. Below, we address your concerns and questions in detail.
>
> **Concerns about strict safety constraints**
>
> Thank you for the comment. We agree that satisfying the condition $ V_c^\pi(s) \leq \kappa $ is essential for a policy $ \pi $ to fully adhere to safety constraints. However, it is worth noting that there may exist a subset of the state space (let us denote it by $ \mathcal{S’}_f $) for which no feasible policy exists. Additionally, this infeasible set may intersect with, or be disjoint from, the state distribution in the offline dataset. In this context, one of the most conservative approaches available in offline safe RL is to learn a policy that minimizes $ V_c^\pi $, which LSPC-S is specifically designed to achieve (regardless of the specific value of $ \kappa $). For any $ \kappa $ and any learned $ V_c(s) $ using the chosen offline RL method, LSPC-S is expected to produce unsafe actions only when conditioned on states within the infeasible set $ \mathcal{S’}_f(\kappa) $. By sampling actions from high-probability regions in the CVAE latent space, LSPC-S provides a conservative policy that reduces the risk of out-of-distribution actions and generates safe actions in expectation.
>
> While LSPC-O similarly samples from this restricted latent space and is theoretically expected to maintain equivalent safety, it may introduce some trade-offs in safety by optimizing for reward-maximizing latent variables rather than random latent samples. Also, we acknowledge that, unlike FISOR [Zheng et al., (2024)],  in this work, we primarily consider a soft constraint for safety, instead of a hard constraint, which can be an extension beyond our current work and of independent interest. We appreciate your feedback very much.
>
>
>
> **Exact sample complexity of the methods**
>
> In this work, we provide an approximate sample complexity for the offline dataset that enables us to achieve a safe and optimal solution. The analysis is built upon the stationary state distribution, as the policy learning is based on a given static dataset, and does not depend on the online interactions with an environment, which often requires us to probe particularly the convergence rate in terms of a dynamic metric such as regret [Zhong et al. (2024)] utilized in the stochastic or online optimization area. Though the direct sample complexity analysis can be significantly beneficial to provide a solid theoretical foundation for the proposed algorithm, a thorough investigation on this aspect is currently out of the scope of our study, and still remains extremely challenging and requires a substantial amount of non-trivial efforts. Notably, a few existing works have reported interesting results in *only* either safe or offline RL settings [Nguyen-Tang et al. (2021); Li et al. (2024); HasanzadeZonuzy et al. (2021); Shi & Chi (2024)], and we have cited them as valuable references for qualitative comparison. However, the approximate sample complexity analysis conducted in our work still offers us meaningful insights on the relationship among the sample size, optimality, and safety, as shown in Theorem 3 and Theorem 4. Additionally, the complexity complies with the existing literature listed above, facilitating the theoretical understanding of offline safe RL.
>
> ***Page (1/4)***

---

> ### Author Response · Authors · 2024-11-20
> **Response to the reviewer He6z's comments (continued)**
>
> **Ablation Study**
>
> Thank you for your constructive comment. We agree that ablation studies are crucial for understanding the impact of individual components of the model on performance. We provide a detailed classification of our hyperparameters in Appendix A.3.1. While ablation studies on IQL and AWR hyperparameters are possible, we would like to note that 1) this line of study has already been explored in depth within the offline RL literature, and such studies generally fall outside the primary scope of our contributions (one could also consider alternative offline RL baselines, such as EQL and XQL, as ablation factors, which would constitute an additional extensive line of investigation); 2) we aimed to keep hyperparameter tuning minimal, opting for standard values that are commonly used across prior offline RL works. Nevertheless, we acknowledge that this particular set of values may not universally apply to every task or dataset.
>
> In our approach, the principal hyperparameter is the restriction parameter $\epsilon$, which we discuss in detail in Appendix A.3.1  with accompanying examples for LSPC-S, LSPC-O and CVAE policy. Additionally, we could perform further ablations on the dimension of the latent safety space. Our chosen dimensionality of 32 seems effective across diverse tasks, likely due to the shared order of dimensionality, which ensures that the latent space is adequately expressive for capturing safety constraints.
> To address the reviewer's interest in ablation studies, we propose the following:
>
> *1. Investigating the effect of reversing the roles of CVAE and safety encoder, where CVAE is trained for reward maximization and the encoder for cost minimization within the inferred latent space.*
>
> To investigate the effect of reversing the roles of the CVAE and safety encoder, we train a model with the architecture depicted here [ https://imgur.com/a/Nh68jCw ], which we refer to as Converse LSPC-O. In this configuration, the CVAE is trained for reward maximization, while the encoder focuses on cost minimization within the inferred latent space.
> We trained Converse LSPC-O for 300k timesteps in the HalfCheetah-velocity environment, varying the restriction level ($\epsilon$) in the latent space. The training curves can be found here  [ https://imgur.com/a/LpGLIRA ], and our key findings include:
>
> a. With low $\epsilon$ (high restriction), the encoder ($\delta$) struggles to learn a cost-reducing policy effectively.
>
> b. As $\epsilon$ increases, we observe a modest decrease in cost, which comes at the expense of lower reward returns.
>
> c. While we can further loosen the restriction to get safer policies, very loose restrictions (high $\epsilon$) may yield out-of-distribution (OOD) actions that falsely appear to minimize cost. This is likely due to the decreasing density of samples seen by the decoder during training as we move away from the mean in the latent space.
>
> d. Importantly, the original LSPC framework generally avoids these issues by prioritizing safety; the samples within the inferred LSPC boundaries are both in-distribution and safe. Additionally, the original framework always maintains a conservative policy (LSPC-S) learned during the training of LSPC-O as a backup.
>
>
> *2. Examining how variations in the inverse temperature hyperparameter affect the performance of LSPC-S and whether this influence extends to LSPC-O.*
>
> In this ablation experiment, the inverse temperature hyperparameter $\lambda$ dictates the trade-off between behavior regularization and value function optimization. For LSPC-S, a lower $\lambda$ indicates a higher degree of behavior cloning, while a larger $\lambda$ places sharper weight on samples with better cost advantages. To evaluate the impact of this parameter on both reward performance and safety adherence, we trained LSPC-S with $\lambda$ values of 1, 2, and 4 in the CarRun and BallCircle environments.
> In these training logs [ https://imgur.com/a/Hl1FehM ], LSPC-S with all three $\lambda$ values maintained safety across both tasks; however, higher $\lambda$ values typically resulted in restricted reward-wise performance. This effect was particularly pronounced in the BallCircle [ https://imgur.com/a/0yvkUmj ] environment, where LSPC-S with $\lambda$=1 significantly outperformed those with $\lambda$=2 and $\lambda$=4. The key question then arises regarding what these varying reward-return levels and similar safety adherence imply for LSPC-O. The results in the figure [ https://imgur.com/a/fRNjjan ] indicate that the reward performance of LSPC-O is enhanced when CVAE (or LSPC-S) is trained with a lower $\lambda$, although this improvement is accompanied by an increase in cost returns. While LSPC-S can achieve high reward returns and balance cost returns with lower $\lambda$ values, the encoder trained on top of it to further maximize reward may lead to unsafe policies.
>
>
> ***Page (2/4)***

---

> ### Author Response · Authors · 2024-11-20
> **Response to the reviewer He6z's comments (continued)**
>
> *3. Analyzing the dependence of our method on the asymmetric loss function of IQL, particularly for cost management and safety adherence.*
>
> We analyzed the impact of varying the coefficient in the asymmetric loss function used for expectile regression in Implicit Q-Learning (IQL) on our LSPC framework. In our original experiments, we used a coefficient of 0.7, which discourages underestimation of cost Q-values. We then compared this to using a coefficient of 0.5, which results in a symmetric Mean Squared Error (MSE) loss, equivalent to SARSA-style policy evaluation. Our findings [ https://imgur.com/a/OzLNuIs ] reveal a trade-off between reward optimization and safety adherence. For LSPC-S, using the symmetric loss ($\xi$=0.5) led to improved reward-wise performance compared to the asymmetric loss ($\xi$=0.7). However, for LSPC-O [ https://imgur.com/a/6yzmTJ1 ], where an additional encoder is trained for reward optimization, using the symmetric loss resulted in higher cost returns, indicating potential safety compromises. These results again highlight the delicate balance between performance and safety in our framework, demonstrating that while less restrictive policies may yield better rewards, they can also lead to increased safety risks when further optimized for performance.
>
>
> We will include this analysis in the revised version of the paper. If you have any suggestions regarding specific ablation studies that you believe would help clarify the paper's contribution, we would greatly appreciate your input.
>
>
>
>
> **Why CVAE among many other generative models**
>
> We acknowledge that more recent generative models have shown promising results in various fields, including reinforcement learning, and may indeed improve performance as the reviewer speculates. However, we emphasize that the choice of generative model is orthogonal to our methodology—rather, the purpose of using such a model is for capturing the safety constraints and reducing distribution shift, which the reviewer also noted in the summary. That said, using certain models like diffusion would require adaptations, as diffusion models typically rely solely on decoders and do not inherently include an encoder component. Thus, using these models together with reward-optimizing encoders might involve further methodological innovation and could be an interesting research direction beyond the current work.
>
> Our decision to use CVAE is largely driven by the relatively low-dimensional nature of the tasks at hand, where CVAE proves expressive enough (as demonstrated by the strong empirical results) and allows for a more interpretable and tractable representation of the inferred latent safety boundary (Figures 1b and 3). We will some technical clarifications into the revised draft.
>
>
>
>
> **Question regarding performance bound on Lemma 2**
>
> We really appreciate this thoughtful comment/question from the reviewer. In Cen’s paper, they have bounded $D_{TV}(d^{\pi}(s)||d^{\*}(s))$ by $D_{TV}(d^{\mathcal{D}}(s)||d^{\*}(s))$ in Lemma 2 based on Assumption 4. This assumption indicates that the stationary state distribution of learned policy is closer to the optimal state distribution than dataset distribution. Therefore, such a replacement makes the bound somewhat looser. Also, in their Theorem 2, they did not provide further analysis on the first term. Instead, in our analysis, we did not use the replacement, but leveraging Lemma 1 to find the relationship between $D_{TV}(d^{\pi}(s)||d^{\*}(s))$ and $D_{TV}(\pi(\cdot|s)||\pi^{\*}(\cdot|s))$ such that we can further proceed the analysis with Lemma 3 in our paper to eventually get a bound with respect to the sample size in Theorem 3 or 4. However, as pointed out by the reviewer, the offline safe RL is more challenging than just offline RL. The tighter bound may be overclaimed in the paper. We agree with the reviewer since the relationship between $D_{TV}(d^{\pi}(s)||d^{\*}(s))$ and $D_{TV}(\pi(\cdot|s)||\pi^{\*}(\cdot|s))$ comes with a cost of $\mathcal{O}(\frac{1}{(1-\gamma)^2})$, instead of $\mathcal{O}(\frac{1}{1-\gamma})$. We will properly state this again in the revised draft and add some necessary clarifications.
>
>
> ***Page (3/4)***

---

> > ### Author Response · Authors · 2024-11-20
> > **Response to the reviewer He6z's comments (continued)**
> >
> > **Comparison of the proposed methods against CPQ**
> >
> > This is a very important question that requires some clarification, especially as CPQ is cited multiple times in our paper. We clarify below:
> >
> > *a. Value Learning:* CPQ uses a cost-sensitive version of CQL, which maximizes the cost value for OOD actions while only using constraint-safe and in-distribution-safe samples to learn the reward Q-function. By contrast, our method does not use a constraint-penalized Bellman operator like CPQ; instead, we adopt IQL for TD learning of both reward and cost values, maintaining standard Bellman updates for simplicity and stability.
> >
> > *b. Policy Extraction:* For policy extraction, CPQ uses the off-policy deterministic policy gradient method, while our approach uses AWR, where the actor is updated via weighted behavior cloning, aligning the policy with the behavior distribution to ensure stability in offline RL settings. Mainly, we train cost-AWR to train the CVAE policy and reward-AWR to train an additional encoder.
> >
> > *c. CVAE usage:* Although both methods incorporate a CVAE, their usage diverges significantly. CPQ employs the CVAE to detect and penalize OOD actions as high-cost in Q-learning, but this distorts the value function and can hinder generalizability, as observed in our experiments and noted in DSRL and FISOR as well [Zheng et al., (2024); Liu et al., (2023)]. Our approach instead uses the CVAE as a policy that expressively models the safety constraints in the latent space, and *not* for an OOD detection metric.
> >
> > We will include these details in the revised paper.
> >
> > **References:**
> >
> > Zheng, Y., Li, J., Yu, D., Yang, Y., Li, S. E., Zhan, X., & Liu, J. (2024). Safe offline reinforcement learning with feasibility-guided diffusion model. arXiv preprint arXiv:2401.10700.
> >
> > Xu, H., Zhan, X., & Zhu, X. (2022). Constraints penalized q-learning for safe offline reinforcement learning. In Proceedings of the AAAI Conference on Artificial Intelligence (Vol. 36, No. 8, pp. 8753-8760).
> >
> > Cen, Z., Liu, Z., Wang, Z., Yao, Y., Lam, H., & Zhao, D. (2024). Learning from sparse offline datasets via conservative density estimation. arXiv preprint arXiv:2401.08819.
> >
> > Nguyen-Tang, T., Gupta, S., Tran-The, H., & Venkatesh, S. (2021). Sample complexity of offline reinforcement learning with deep ReLU networks. arXiv preprint arXiv:2103.06671.
> >
> > Li, G., Shi, L., Chen, Y., Chi, Y., & Wei, Y. (2024). Settling the sample complexity of model-based offline reinforcement learning. The Annals of Statistics, 52(1), 233-260.
> >
> > HasanzadeZonuzy, A., Bura, A., Kalathil, D., & Shakkottai, S. (2021, May). Learning with safety constraints: Sample complexity of reinforcement learning for constrained mdps. In Proceedings of the AAAI Conference on Artificial Intelligence (Vol. 35, No. 9, pp. 7667-7674).
> >
> > Shi, L., & Chi, Y. (2024). Distributionally robust model-based offline reinforcement learning with near-optimal sample complexity. Journal of Machine Learning Research, 25(200), 1-91.
> >
> > Zhong, H., & Zhang, T. (2024). A theoretical analysis of optimistic proximal policy optimization in linear markov decision processes. Advances in Neural Information Processing Systems, 36.
> >
> > Liu, Z., Guo, Z., Lin, H., Yao, Y., Zhu, J., Cen, Z., ... & Zhao, D. (2023). Datasets and benchmarks for offline safe reinforcement learning. arXiv preprint arXiv:2306.09303.
> >
> > ***Page (4/4)***

---

> > > ### Comment · Reviewer_He6z · 2024-11-25
> > >
> > > Sorry for the delayed response. Your answer addressed my main concerns. However, I still have reservations about the safety of the proposed method. In my view, the safety guarantee of this work stems from two aspects: (1) the cost-value function learned by IQL; and (2) the CVAE trained on the dataset. If the dataset contains unsafe actions and states, the CVAE cannot guarantee safety, especially for LSPC-O. If you can demonstrate the safety of LSPC-O under a dataset that includes both unsafe and safe samples, I will raise my score.
> > >
> > > By the way, I think the connection between your method and theoretical analysis is not very strong. If you can more clearly illustrate the relationship between LSPC and the theoretical analysis, it would improve your work.

---

> > > > ### Author Response · Authors · 2024-11-26
> > > >
> > > > **Safety of LSPC-O under dataset that includes both unsafe and safe samples**
> > > >
> > > > Thank you for highlighting your concern regarding the safety of LSPC-O when applied to datasets containing both safe and unsafe samples. We would like to clarify that almost all of the datasets used in our evaluations include both safe and unsafe trajectories. Please note that the safety compliance of policies during evaluation is assessed based on their adherence to the undiscounted cost return threshold established in our experiments (consistent with standard benchmarks). The classification of safe and unsafe trajectories in the dataset follows the same criteria.
> > > >
> > > > To further address your concern, we have created scatter plots illustrating the reward return versus cost return for trajectories in some datasets, including the ones used in the ablation study above. The red dashed lines represent the cost return thresholds. Each point in these plots represents a trajectory/episode, providing a visual depiction of the dataset's composition. Additionally, on the right, we include frequency distributions of the cost returns for these datasets, which demonstrate that many trajectories exceed the safety threshold, often with unsafe trajectories outnumbering safe ones. These visualizations reinforce the robustness of LSPC-O in handling datasets with mixed safety profiles. We hope this additional evidence alleviates your concerns regarding the empirical safety of the policy learned by our method.
> > > >
> > > > Link to the plots: [ https://imgur.com/a/k0axTpW ]
> > > >
> > > > **Relationship between LSPC and the theoretical analysis**
> > > >
> > > > Thank you for your suggestion, we really appreciate it. We would like to take this opportunity to further clarify this relationship.
> > > >
> > > > In LSPC, we leverage CVAE to reconstruct safe policies residing in the offline dataset $\mathcal{D}$, strategically ensuring that the further optimal solutions are safe. This justifies Assumption 1 and Assumption 2 that provide new upper bounds for the distribution distances between $\pi$ and $\pi_s$, and $\pi_s$ and $\pi^*$. Previously, instead of $\pi_s$, it was the unknown behavior policy $\pi_b$ [Yao, et al. 2024], which has less assurance of safety than $\pi_s$ and results in relatively looser bounds. Moreover, in LSPC, we adopt the IQL and AWR for critic model updates and policy extraction, which motivates us to use the value function as the metric to evaluate the proposed algorithm. As the policy is learned via a static offline dataset, the performance gap and constraint violation are assessed based on the stationary distributions, instead of a dynamic metric such as regret [Zhong and Zhang 2024]. Theorem 1 and Theorem 2 suggest the performance gap and constraint violation during learning are upper bounded by constants that are correlated with the new distribution distance bounds induced by the safety policy $\pi_s$. Additionally, as the policy evaluation and extraction steps are all parametric with deep learning models, the number of data samples plays a central role in controlling the performance, which has been a well-established fact in modern deep learning community.
> > > > In offline RL, this motivates the need to assess performance gap and constraint violations with respect to the size of $\mathcal{D}$. Thereby, Theorem 3 and Theorem 4 imply the decay rate with respect to the size of $\mathcal{D}$. Though the analysis can apply to related algorithms with some adaptation, the constants in these results, particularly $\varepsilon_1'$ and $\varepsilon_2'$, are uniquely defined, and thus the theoretical conclusions are tailored to the proposed algorithm.
> > > >
> > > > We are working on the revised manuscript accommodating the major comments from all the reviewers and we will make sure to include these recent discussions as well.
> > > >
> > > > **References**
> > > >
> > > > Yao, Y., Cen, Z., Ding, W., Lin, H., Liu, S., Zhang, T., ... & Zhao, D. (2024). Oasis: Conditional distribution shaping for offline safe reinforcement learning. arXiv preprint arXiv:2407.14653.
> > > >
> > > > Zhong, H., & Zhang, T. (2024). A theoretical analysis of optimistic proximal policy optimization in linear markov decision processes. Advances in Neural Information Processing Systems, 36.

---

> > > > > ### Comment · Reviewer_He6z · 2024-11-27
> > > > >
> > > > > Thank you to the authors for addressing most of my concerns. However, I believe the analysis in this paper seems somewhat like a generic proof that could apply to any method. It would be better to demonstrate that the KL gap for IQL is narrower than that of CQL. For example, if we use IQL to learn $V$, the $\epsilon$ in Assumptions 1-4 might be smaller than for CQL. Nevertheless, due to the author’s hard work, I will increase my score.

---

> > > > > > ### Author Response · Authors · 2024-11-27
> > > > > >
> > > > > > We appreciate your feedback, agree with your observation and acknowledge the need for clarification regarding this point.
> > > > > >
> > > > > > As noted in our previous reply, the analysis framework could indeed be adapted to related algorithms. However, the constants and results in our analysis are specific to the proposed method. For instance, the constrained reward-return maximization (equations 12–14) is central to our policy extraction, as it allows us to leverage the assumption that the KL divergence between \pi and \pi_s is upper bounded by $\varepsilon’_1$. This assumption later enables the derivation of performance bounds. More clearly, our contribution lies in deriving $\pi_s$ within the distribution governed by the behavior policy $\pi_b$ and subsequently extracting the optimal policy approximation $\pi$ within the constraints of $\pi_s$​. The theoretical analysis aligns with this structure, focusing on constrained reward-return maximization to derive performance bounds specific to our approach. Naturally, as you noted, future works in safe offline RL could build upon these analyses to derive performance bounds for their methods, provided they employ similar policy extraction strategies and justify a comparable set of assumptions.
> > > > > >
> > > > > > We also appreciate the suggestion regarding the KL gap. We would like to clarify that demonstrating discrepancies in KL gaps between policies due to the choice of value learning method is not directly related to our analysis. We chose to use IQL in our experiments because it worked well for our needs. However, our main contribution does not depend on which value learning method is used, so testing other methods could be an interesting direction for future research. We will include this point as a future direction in the paper.
> > > > > >
> > > > > > We hope this addresses the concern and provides a clear explanation of the scope and focus of our contribution.

---

### Author Response · Authors · 2024-11-27
**Manuscript Update**

We sincerely thank all the reviewers for their thorough and insightful feedback on our manuscript. In response to the reviewers’ comments and the discussions that followed, we have revised our manuscript accordingly and are uploading the updated version for your consideration. Below, we summarize the key modifications and additions along with their respective sections or locations in the revised document.

Major changes/additions to the manuscript:
1. Added Ablation Studies (A.3.2)
2. Softened Claims and Clarification surrounding Lemma 2 (Section 4, Lemma 2)
3. Added comparison between LSPC and CPQ (A.3.5)
4. Additional Remark 2 for clarification on theoretical analysis (Section 4)
5. Added discussion on the safety profile of the dataset (A.5)
6. Added discussion on safety-performance trade-offs in LSPC-S and LSPC-O (A.3.3)

Minor changes/additions:
1. References to respective Github repositories for baseline implementation and hyperparameter configuration (A.4)
2. References to the GitHub repositories for each task set (A.4.1-A.4.3)
3. Computational time (A.3.4)
4. Changed table formatting and fonts used in the figure for better readability (Tables 1-6, Figures 2-3)

Due to space constraints, most of these updates had to be placed in the appendix. However, following upcoming discussions with the reviewers, we remain open to further revisions. Depending on the importance of specific results or theoretical insights, we are prepared to incorporate them into the main text.

---

### Meta-Review · Area_Chair_7TL5 · 2024-12-21

**Metareview:**

This paper studies safe offline reinforcement learning by proposing a Latent Safety-Constrained Policy approach, employing Conditional Variational Autoencoders (CVAE) to model latent safety constraints and optimizing policies within this framework. It presents theoretical bounds and empirical results demonstrating improved safety and reward performance compared to existing methods. However, the reviewers raised concerns about the weak connection between the theoretical analysis and the practical algorithm, particularly in linking the theoretical results to the method's performance. Overall the paper is on the borderline.

**Additional Comments On Reviewer Discussion:**

During the rebuttal period, the reviewers raised several points about the submission. Reviewer He6z expressed concerns about the weak connection between the theory and the algorithm, particularly regarding safety guarantees and sample complexity, which the authors addressed partially by adding clarifications and ablation studies but did not fully resolve to the reviewer’s satisfaction. Reviewer NRFb noted challenges in connecting the theoretical analysis to the method and questioned the framework's generalization to real-world scenarios; the authors clarified theoretical assumptions but acknowledged limitations, which the reviewer found adequate but not convincing enough to raise their score. Reviewer Qh7M highlighted the potential limitations of the KL divergence-based safety assurance and oscillations in cost during training; the authors explained these aspects and provided additional empirical insights, which were deemed partially satisfactory. Reviewer MBPj raised questions about the reward-cost trade-off and oscillations in LSPC-O; the authors offered detailed explanations and additional computation details, which the reviewer appreciated but saw as requiring further exploration.

---

### Decision · Program_Chairs · 2025-01-22

Accept (Poster)